*Version of August 26, 2024*

# The impact of dehydration and extremely low HCl values in the Antarctic stratospheric vortex in mid-winter on ozone loss in spring

Yiran Zhang-Liu[1], Rolf Müller[1,5], Jens-Uwe Grooß[1,5], Sabine Robrecht[1,2], Bärbel Vogel[1,5], Abdul Mannan Zafar[1,3], and Ralph Lehmann[4]

[1]Institute of Climate and Energy Systems (ICE-4), Forschungszentrum Jülich, Jülich, Germany
[2]Deutscher Wetterdienst, Offenbach, Germany
[3]Biotechnology Research Center, Technology Innovation Institute, Masdar City, Abu Dhabi, United Arab Emirates
[4]Alfred Wegener Institute, Helmholtz Centre for Polar and Marine Research, Potsdam, Germany
[5]Center for Advanced Simulation and Analytics (CASA), Forschungszentrum Jülich, Jüich, Germany

**Correspondence:** Rolf Müller (ro.mueller@fz-juelich.de) and Yiran Zhang-Liu (yiranz.luna@gmail.com)

**Abstract.** Simulations of Antarctic chlorine and ozone chemistry in previous work show that in the core of the Antarctic vortex (16–18 km, 85–55 hPa, 390–430 K) HCl null cycles (initiated by reactions of Cl with $CH_4$ and $CH_2O$) are effective. These HCl null cycles cause both HCl molar mixing ratios to remain very low throughout Antarctic winter/spring and ozone destroying chlorine ($ClO_x$) to remain enhanced, so that rapid ozone depletion proceeds. Here we investigate the impact of the observed dehydration in Antarctica, which strongly reduces ice formation and the uptake of $HNO_3$ from the gas phase; however the efficacy of HCl null cycles is not affected. Moreover, also when using the observed very low HCl molar mixing ratios in Antarctic winter as initial value; HCl null cycles are efficient in maintaining low HCl (and high $ClO_x$) throughout winter/spring. Further, the reaction $CH_3O_2 + ClO$ is important for the efficacy of the HCl null cycle initiated by the reaction $CH_4 + Cl$. Using the current kinetic recommendations instead of earlier ones has very little impact on the simulations. All simulations presented here for the core of the Antarctic vortex show extremely low minimum ozone values (below 50 ppb) in late September/early October in agreement with observations.

## 1 Introduction

The Antarctic ozone hole is a phenomenon of substantially reduced polar ozone that has reoccurred every winter and spring over Antarctica for about four decades (Jones and Shanklin, 1995; Müller et al., 2008; Bodeker and Kremser, 2021; WMO, 2022; Klekociuk et al., 2022; Johnson et al., 2023; Roy et al., 2024). In many years, the Antarctic ozone hole shows very low ozone in the altitude region of 14-21 km ($\sim$ 380-550 K potential temperature) (Solomon et al., 2005; Jurkat et al., 2017; Johnson et al., 2023). In exceptional years, sudden stratospheric warmings occur in the Antarctic (2002 and 2019) causing unusually low ozone depletion (e.g., Müller et al., 2008; Grooß et al., 2005; Smale et al., 2021). Substantial polar ozone loss

can also occur in the Arctic; albeit the ozone loss shows a much stronger year-to-year variability (e.g., Müller et al., 2008; Johansson et al., 2019; Wohltmann et al., 2020; Dameris et al., 2021; Grooß and Müller, 2021; von der Gathen et al., 2021; Ardra et al., 2022). Polar ozone depletion is ultimately driven by chlorine and bromine substances released to the atmosphere as a result of human activities. Notwithstanding there are also bromine and chlorine substances with natural sources (e.g.,
WMO, 2022; Lauther et al., 2022; Jesswein et al., 2022). The release of the human made chlorine and bromine substances to the atmosphere has led to a substantial increase in the atmospheric halogen loading in the latter half of the last century; human made halogen compounds started increasing substantially in the atmosphere since the 1960s. Consequently, with the stratospheric halogen loading declining since about 2000, the first signs of recovery of both the Antarctic ozone hole and global ozone levels are observed (e.g., Várai et al., 2015; Kuttippurath and Nair, 2017; Strahan and Douglass, 2018; WMO, 2022;
Bodeker and Kremser, 2021; Stone et al., 2021; Weber et al., 2022; Johnson et al., 2023).

For a successful simulation of the total column ozone field in the Antarctic vortex in spring, both the stratospheric chlorine and bromine chemistry need to be represented correctly in a model as well as the dynamical isolation of the Antarctic vortex (e.g., Sonnabend et al., 2024). Current chemistry-climate models allow many characteristics of the global total column ozone field to be reproduced, but there is a considerable spread among models in the predictions of the absolute ozone column and the
simulation of the Antarctic ozone hole is often not satisfactory. Possible reasons could be deficiencies in the model dynamics or in the stratospheric chemistry scheme of chemistry-climate models (Struthers et al., 2009; Dhomse et al., 2018). Such issues also impact the reliability and the accuracy of projections of the recovery of the Antarctic ozone hole under different climate scenarios for the future including climate intervention (Jöckel et al., 2016; Dhomse et al., 2018; Tilmes et al., 2021).

For polar stratospheric ozone depletion to occur, chlorine (which mostly prevails in the stratosphere in the form of the
reservoir species HCl and $ClONO_2$) needs to be converted to an ozone destroying form. That is, HCl and $ClONO_2$ need to be "activated" by heterogeneous reactions on polar stratospheric clouds (PSCs) or cold sulphate aerosol particles (Portmann et al., 1996; Solomon, 1999; Shi et al., 2001; Drdla and Müller, 2012; WMO, 2022; Tritscher et al., 2021). Ozone depletion occurs with the return of sunlight to the polar region; this time period is characterised by maintenance of high levels of active chlorine (e.g., Santee et al., 2005; Santee et al., 2008; Solomon et al., 2015; Nedoluha et al., 2016; Jurkat et al., 2017; Wohltmann et al.,
2017; Müller et al., 2018; Johansson et al., 2019; Roy et al., 2024). When PSCs occur, $HNO_3$ is sequestered in PSC particles and thus removed from the gas-phase. If the $HNO_3$ containing PSC particles sediment, permanent denitrification in the polar stratosphere occurs (e.g., de Laat et al., 2024). PSCs are present in the Antarctic lower stratosphere throughout winter until early October, whereas in the Arctic PSCs occur with much greater year-to-year variability (Pitts et al., 2009; Spang et al., 2018).

In the initial step of chlorine activation, in the heterogeneous reaction

$$HCl + ClONO_2 \rightarrow Cl_2 + HNO_3 \tag{R1}$$

the available $ClONO_2$ is titrated against HCl (e.g., Solomon et al., 1986; Wohltmann et al., 2017). In the Antarctic lower stratosphere, the initial concentrations of HCl are greater than those of $ClONO_2$ (Jaeglé et al., 1997; Santee et al., 2008; Nakajima et al., 2020). Thus, in the absence of chemical processes leading to a further loss in HCl, there is no full activation in

this step. Such a behaviour is found in models (Grooß et al., 2018). Then a period of relatively little chemical change in polar night follows ("sleeping chemistry").

In austral spring, in the core of the Antarctic vortex at altitudes of 16–18 km (85–55 hPa, 390–430 K), high values of active chlorine ($ClO_x = Cl + ClO + 2 \times Cl_2O_2$) are maintained in spite of increasingly rapid formation of HCl in the gas phase through reactions of Cl with $CH_4$ and $CH_2O$ (Müller et al., 2018). During this period the most rapid ozone depletion occurs. For such conditions, the maintenance of high $ClO_x$ values is accomplished by effective reaction cycles ("HCl null cycles") in which deactivation (i.e. production of HCl) is immediately balanced by the heterogeneous reaction of HCl with HOCl

$$HCl + HOCl \rightarrow Cl_2 + H_2O \tag{R2}$$

(Crutzen et al., 1992; Prather, 1992), which occurs on the surfaces of nitric acid trihydrate (NAT) and ice particles or within supercooled (liquid) ternary solutions and cold liquid aerosol particles. Further, the reaction

$$ClO + CH_3O_2 \rightarrow Cl + CH_3O + O_2 \tag{R3}$$

is essential for the HCl null cycle initiated by the reaction of Cl with $CH_4$ (Crutzen et al., 1992; Zafar et al., 2018, see also AR1-AR8 in appendix A).

However, at altitudes somewhat greater than 18 km (55 hPa, 430 K) and for conditions in the lower stratosphere closer to the edge of the polar vortex, $HNO_3$ will not continuously be sequestered in PSCs, so that periods with enhanced gas-phase concentrations of $HNO_3$ (compared to the vortex core) will occur. Under such conditions, more $NO_2$ will be available in the gas-phase (e.g., de Laat et al., 2024), enhancing the production of $ClONO_2$, so that reaction R1 will have a stronger impact on chlorine chemistry. As a result, the chemistry of HCl null cycles will be more complex.

The period of strongly enhanced $ClO_x$ and strong ozone loss in the Antarctic ends with a very rapid formation of HCl leading to a practically complete conversion of $ClO_x$ to HCl through the reactions of Cl with $CH_4$ and $CH_2O$ (i.e., to deactivation) (e.g., Crutzen et al., 1992; Douglass et al., 1995; Grooß et al., 1997; Grooß et al., 2011; Nakajima et al., 2020). However, very rarely, when the Antarctic vortex is perturbed by a sudden stratospheric warming (2002 and 2019), there is less ozone depletion and significant deactivation into $ClONO_2$ may also occur (Grooß et al., 2005; Smale et al., 2021).

Heterogeneous chlorine activation, enhanced concentrations of active chlorine and subsequent ozone loss occur frequently in the polar regions. Under exceptional circumstances chlorine activation also occurs in the mid-latitudes for conditions of low temperatures and enhanced water vapour. The surfaces for heterogeneous reactions might be provided for example by stratospheric ice particles, stratospheric sulphate aerosol particles (potentially enhanced by volcanic eruptions or climate intervention) or by wildfire smoke injected into the stratosphere (e.g. Solomon et al., 1997; Tilmes et al., 2008; von Hobe et al., 2011; Klobas et al., 2017; Robrecht et al., 2019, 2021; Tilmes et al., 2021; Ohneiser et al., 2022; Santee et al., 2022).

In the present study, we perform sensitivity analyses, exploring the influence of different parameters on the rate of these HCl null cycles and the resulting ozone loss. We extend earlier work (Grooß et al., 2011; Müller et al., 2018; Zafar et al., 2018) investigating the chemical processes in the core of the Antarctic vortex in the lower stratosphere (16–18 km, 85–55 hPa, 390–430 K), where extremely low ozone molar mixing ratios in spring are reached regularly (Solomon et al., 2005; Johnson

et al., 2023). (Molar mixing ratios are identical to volume mixing ratios in the case of an ideal gas). As in the earlier work (Grooß et al., 2011; Müller et al., 2018), we rely on a detailed examination of a single trajectory and an analysis of multi-

trajectory simulations. Here we do *not* employ a three-dimensional model version (see also section 2.1 below), which is based on global or hemispheric meteorological fields and includes atmospheric mixing (e.g., Poshyvailo et al., 2018; Grooß and Müller, 2021; Sonnabend et al., 2024). We now use the most recent recommendation (Burkholder et al., 2020) of chemical kinetics and photochemical data.

In particular, we take into account the impact of the observed Antarctic dehydration (e.g., Kelly et al., 1989; Vömel et al.,

1995; Nedoluha et al., 2002; Jiménez et al., 2006; Ivanova et al., 2008; Rolf et al., 2015), which was not propperly taken into account in earlier work (Müller et al., 2018; Zafar et al., 2018). Further, the impact of very low HCl molar mixing ratios in Antarctic winter (Wohltmann et al., 2017; Grooß et al., 2018) is now considered. Both dehydration and very low HCl molar mixing ratios are clearly observed in the atmosphere.

Taking into account the observed dehydration in the Antarctic vortex (see also section 2.2.1 below for details) reduces

substantially the occurrence of ice clouds in the model. Ice clouds are very efficient in sequestering $HNO_3$ from the gas-phase (e.g. Hynes et al., 2002), thus a lower occurrence of ice clouds in the model reduces substantially the uptake of gas-phase $HNO_3$ on ice particles; however we find that the efficacy of HCl null cycles is not affected. Assuming an HCl mixing ratio of zero after polar night while keeping total inorganic chlorine ($Cl_y$) constant (Wohltmann et al., 2017; Grooß et al., 2018, see also section 2.2.2 below for details) approximately takes into account the observed very low HCl molar mixing ratios in the

Antarctic vortex in mid-winter. This assumption leads to very low HCl molar mixing ratios throughout Antarctic winter and spring, but the efficacy of HCl null cycles is again not affected.

The simulations presented here show that neither of these two assumptions (dehydration and very low HCl molar mixing ratios) has a very strong effect on the simulated chemical ozone depletion compared to earlier work (Müller et al., 2018; Zafar et al., 2018); similarly, using the most recent recommendation (Burkholder et al., 2020) has very little impact. Severe ozone

depletion to values below 50 ppb is simulated (consistent with observations) for the South Pole in late September and early October.

In summary, our box-model calculations of Antarctic chlorine and ozone chemistry corroborate earlier findings that HCl null cycles, in the core of the vortex in the lower stratosphere in spring are effective in allowing high levels of active chlorine to be maintained and rapid ozone loss to proceed. We show here that these conclusions are not changed when current kinetic

recommendations (Burkholder et al., 2020) are employed or when dehydration and very low HCl molar mixing ratios, both observed in polar winter, are taken into account.

## 2 Methods

### 2.1 Chemical model

The simulations reported here were performed with the Chemical Lagrangian model of the Stratosphere (CLaMS, McKenna

et al., 2002; Grooß et al., 2005, 2018) with a set-up following closely the one used earlier (Grooß et al., 2011; Müller et al., 2018;

Zafar et al., 2018). Briefly, here, the stratospheric chemistry is calculated for particular air parcels along three-dimensional trajectories. To integrate the differential equations representing the set of chemical reactions considered here, we use the solver SVODE (Brown et al., 1989). Chemical rate constants and photolysis cross sections generally are taken from the most current recommendation (Burkholder et al., 2020), but the earlier recommendations by Sander et al. (2011) were used for comparison. Photolysis rates are calculated in spherical geometry (Becker et al., 2000).

Heterogeneous chemistry in the model is assumed to occur on the surface of ice and NAT (with a particle density of $3 \cdot 10^{-3}\,\mathrm{cm}^{-3}$), as well as in supercooled liquid ternary particles ($HNO_3/H_2SO_4/H_2O$) and cold liquid binary ($H_2SO_4/H_2O$) particles. The occurrence of particles in the model is determined by the temperature of the air mass. NAT particles are assumed to form at a supersaturation of 10 from liquid ternary solutions or from ice evaporation. Ice is formed in the model at the equilibrium temperature (no supersaturation). The initial density of liquid (binary) aerosol particles is assumed to be $10\,\mathrm{cm}^{-3}$. The condensable material for liquid ternary particles, NAT and ice is determined from the equilibrium with the gas-phase. The temperature dependent reaction probabilities in liquid ternary particles ($HNO_3/H_2SO_4/H_2O$) and cold liquid binary ($H_2SO_4/H_2O$) particles are determined from recent recommendations (Burkholder et al., 2020). See also Grooß et al. (2011) and Müller et al. (2018) for further details.

## 2.2 Trajectory and chemical set-up

Here trajectories for the austral winter 2003 are used, which are defined by the location and time of minimum ozone measured by ozone sondes at the South Pole. Forward and backward trajectories are calculated from the South Pole at different days (Grooß et al., 2011; Müller et al., 2018). We focus on one particular trajectory passing through the location of an ozone sonde measurement at South Pole; 14 ppb $O_3$ at 74 hPa (391 K) on 24 September 2003 (Grooß et al., 2011). The same trajectory was investigated in earlier work (Müller et al., 2018; Zafar et al., 2018) and is referred to below as the reference trajectory (see also section 3.3.1).

Meteorological data were taken from operational analyses of the European Centre for Medium-range Weather Forecasts (ECMWF) and diabatic descent rates of the air-parcels were calculated using a radiation code (Zhong and Haigh, 1995) and climatological ozone and water vapour profiles (Grooß and Russell, 2005).

The initial molar mixing ratios for the reference trajectory for the main trace gases on 1 June are listed in Table 1; with the exception of $H_2O$ they are the same as in earlier work (Müller et al., 2018; Zafar et al., 2018). The initial value for $ClONO_2$ is extremely low. These initial conditions imply the assumption that for the air parcels in question, the initial step of heterogeneous chlorine activation (reaction R1) has already occurred, so that $ClO_x$ values are enhanced and the HCl molar mixing ratio is lower than at the beginning of the winter. Further, Antarctic denitrification (e.g., de Laat et al., 2024) is also represented in the initial conditions by assuming $HNO_3 = 4.5$ ppb. The sensitivity of the results of the simulations to the initial ozone and initial $HNO_3$ molar mixing ratios, as well as the impact of assumptions on the chemistry of methylhypochlorite ($CH_3OCl$) and the methyl peroxy radical ($CH_3O_2$) has been discussed in previous work (Müller et al., 2018; Zafar et al., 2018).

| | |
|---|---|
| $O_3$ | 2.2 ppm |
| $H_2O$ | 2.05 ppm |
| $CH_4$ | 1.2 ppm |
| $HNO_3$ | 4.5 ppb |
| HCl | 1.05 ppb |
| $ClO_x$ | 1.01 ppb |
| $ClONO_2$ | 12 ppt |
| HOCl | 4.65 ppt |
| $Br_y$ | 17 ppt |
| CO | 16 ppb |

**Table 1.** Initial molar mixing ratios (for 1 June) of atmospheric trace gases used for the CLaMS simulation along the reference trajectory.

### 2.2.1  Initial water vapour

There is one important exception to the initial values used previously, namely the initial value of water vapour. Assuming $H_2O = 4.1$ ppm (Müller et al., 2018; Zafar et al., 2018) is an appropriate estimate for 1 June; however such a value means essentially that irreversible dehydration which occurs thereafter through ice particle sedimentation is neglected. Dehydration occurs every year in the Antarctic, with the removal of water vapour from the air at sufficiently low (ice-formation) temperatures (e.g., Jiménez et al., 2006; Tritscher et al., 2019, 2021). Particles of different sizes will sediment at different rates (Müller and Peter, 1992). Further, there is a year-to-year variability in the extent and timing of the severity of Antarctic dehydration (e.g., Nedoluha et al., 2002) and the dehydration is not uniform throughout the Antarctic vortex (Kelly et al., 1989; Ivanova et al., 2008). Nonetheless, strong dehydration in the Antarctic winter vortex has been reported consistently both for in-situ and remote sensing measurements as well as for model simulations (Kelly et al., 1989; Vömel et al., 1995; Nedoluha et al., 2002; Jiménez et al., 2006; Ivanova et al., 2008; Schoeberl and Dessler, 2011; Rolf et al., 2015; Poshyvailo et al., 2018; Tritscher et al., 2019, 2021).

Kelly et al. (1989) report minimum values of $H_2O$ based on aircraft measurements down to 1.5 ppm (at about $\approx 350$ K) and $H_2O$ molar mixing ratios of 2.0-2.4 ppm for an isentropic flight on 430 K on 2 September 1987. Kelly et al. (1989) also report temperatures corresponding to ice saturation molar mixing ratios of $\sim 2$ ppm over an altitude range 350-450 K in late August at the South Pole. Vömel et al. (1995) conducted a series of balloon measurements with a frost point hygrometer from McMurdo station (in 1990) and South Pole (1990-1994) and find an average molar mixing ratio of $H_2O$ of 2.3 ppm between 16 and 18 km. Ivanova et al. (2008) report airborne $H_2O$ measurements in the core of the Antarctic vortex on 21 and 23 September and 2 and 8 October 1999 during the APE-GAIA campaign – measurements at 410-430 K cover the range between 1.8-2.5 ppm molar mixing ratio of $H_2O$. Jiménez et al. (2006), based on observations by the Microwave Limb Sounder (MLS) report reductions of $H_2O$ up to $\sim 3$ ppm – they find $H_2O$ molar mixing ratios at $\sim 17$ km (440 K) at about 80°S equivalent latitude on 15 September of 2-2.2 ppm water vapour.

Here (for about 430 K) we assume an initial molar mixing ratio of $H_2O = 2.05$ ppm, which accounts for the observed dehydration in Antarctica during winter; 2.05 ppm of water vapour will remain in the gas-phase if a temperature of $\sim 185$ K is reached in the atmosphere. A value for initial $H_2O = 2.05$ is also close to the value of initial $H_2O = 2.2$ ppm employed in earlier work (Crutzen et al., 1992) and is appropriate for conditions in late winter and early spring in the Antarctic.

### 2.2.2   Initial HCl

For Antarctic HCl, a discrepancy between simulations and observations between May and July was reported (Wohltmann et al., 2017; Grooß et al., 2018); during this period model simulations significantly overestimate the observed HCl molar mixing ratios. The process causing this discrepancy between observations and simulations is not known at this point in time. Different options include an increased uptake of HCl into PSC particles (Wohltmann et al., 2017), a temperature bias of the underlying meteorological analyses, and unknown (possibly heterogeneous) chemical reactions (Grooß et al., 2018).

Here we do not investigate a process missing in models (Wohltmann et al., 2017; Grooß et al., 2018); rather our choice of the initial conditions is aimed at estimating the maximum possible effect of the very low HCl molar mixing ratios on HCl null cycles, $ClO_x$, and ozone depletion. This is the case when assuming initial HCl to be zero, with a corresponding increase in $ClO_x$, i.e., initial $Cl_y$ is unchanged in the simulations.

### 2.3   Kinetic and photochemical parameters

| Simulation | Initial HCl | Initial $H_2O$ | Kinetics (JPL recomm.) | Colour |
|---|---|---|---|---|
| S1 | HCl = 1.05 ppb | $H_2O$ = 4.1 ppm | (Sander et al., 2011) | magenta |
| S2 | HCl = 1.05 ppb | $H_2O$ = 4.1 ppm | (Burkholder et al., 2020) | ochre |
| S3 | HCl = 1.05 ppb | $H_2O$ = 2.05 ppm | (Burkholder et al., 2020) | blue |
| S4 | HCl = 0.0 ppb | $H_2O$ = 2.05 ppm | (Burkholder et al., 2020) | red |

**Table 2.** Employed assumptions for four different box-model simulations (S1-S4) a long the reference trajectory. The colours refer to those used in Figs. 2, 4, and 5 below.

In the results presented below, the sensitivity of Antarctic chlorine chemistry and ozone loss in spring to dehydration, to low early winter HCl molar mixing ratios and to different recommendations for kinetic parameters (Sander et al., 2011; Burkholder et al., 2020) is explored. The different model simulations (S1-S4) and the employed assumptions (as well as the colours used in the figures below) are summarised in Table 2.

### 2.3.1   Recent recommendations of chemical kinetic and photochemical data

Müller et al. (2018) and Zafar et al. (2018) presented results of box-model simulations for the lowermost stratosphere in the core of the Antarctic vortex based on earlier recommendations of chemical kinetic and photochemical data by the Jet Propulsion

Laboratory (JPL, Sander et al., 2011, simulation S1 in Table 2). Simulation S1 is shown here to establish a link to earlier work. Otherwise (simulations S2-S4) we use here chemical kinetics and photochemical data from the most recent recommendation (Burkholder et al., 2020).

The observed partitioning of ClO and $Cl_2O_2$ in the Antarctic stratosphere is well represented when the recommendations by Burkholder et al. (2015) are used (Canty et al., 2016). As the rate constants affecting chlorine chemistry were not changed substantially between the recommendations by Burkholder et al. (2015) and Burkholder et al. (2020), polar chlorine chemistry is also well represented by the most recent recommendation (see also appendix B). The largest remaining uncertainties in the ozone depletion rate are caused by uncertainties in the kinetic parameters influencing the $Cl_2O_2$ photolysis rate (Kawa et al.,

2009; Canty et al., 2016; Wohltmann et al., 2017).

### 2.3.2    The rate constant of the reaction ClO + CH$_3$O$_2$

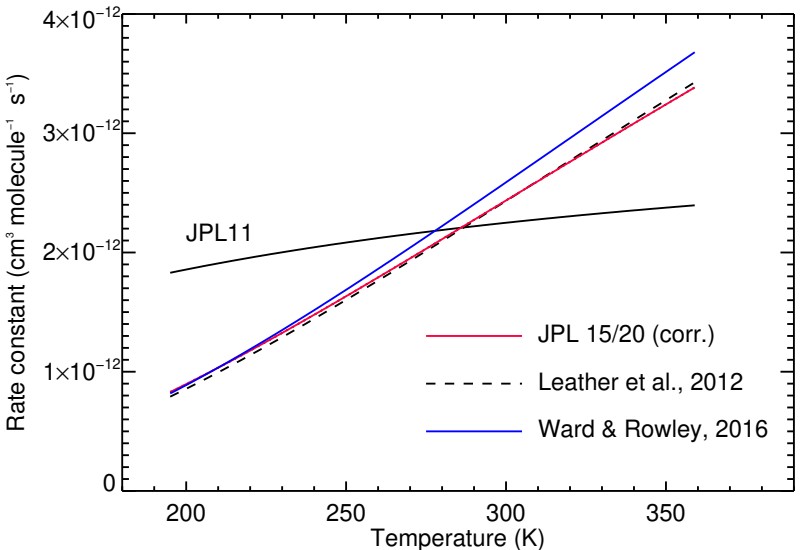

**Figure 1.** The temperature dependent rate constant (in $cm^3$ molecule$^{-1}$ s$^{-1}$) for reaction R3 ($ClO + CH_3O_2 \longrightarrow$ products) from a variety of sources. The recommendation by Sander et al. (2011) is shown as a black solid line (JPL 11) and the red line (JPL 15/20) is for Burkholder et al. (2015) (and Burkholder et al., 2020) with the corrected value $A = 1.8 \times 10^{-11}$ $cm^3$ molecule$^{-1}$ s$^{-1}$. Also shown are recent measurements (dashed line, Leather et al., 2012); (blue line, Ward and Rowley, 2016).

The temperature dependence of the rates of bimolecular reactions are given by the Arrhenius equation

$$k = A \cdot \exp\left(-\frac{E_a}{R \cdot T}\right) \tag{1}$$

here $k$ is the rate constant (in $cm^3$ molecule$^{-1}$ s$^{-1}$), $A$ is a pre-exponential factor (with the same unit as $k$), $T$ is temperature

(in K), $R$ is the universal gas-constant, and $E_a$ may be interpreted as the molar activation energy of the reaction. In recommendations (Burkholder et al., 2015, 2020), commonly values for $A$ (in $cm^3$ molecule$^{-1}$ s$^{-1}$) and $E_a/R$ (in K) are listed. In the

simulations reported below, we corrected the rate constant of reaction R3 ($ClO + CH_3O_2 \rightarrow$ prod.) compared to the listing in recommendations (Burkholder et al., 2015, 2020); the $A$-factor for the rate constant of reaction R3 is

$$A = 1.8 \times 10^{-11} \quad \text{and not} \quad A = 1.8 \times 10^{-12}; \tag{2}$$

the latter value is listed incorrectly (a typo; J. Burkholder, pers. comm.) both in Burkholder et al. (2015) and Burkholder et al. (2020). The temperature dependent rate constant is listed in Sander et al. (2011) as:

$$k_{\text{jpl11}}(T) = 3.3 \times 10^{-12} \cdot \exp\left(-\frac{115}{T}\right) \quad . \tag{3}$$

This recommendation was updated (Burkholder et al., 2015, 2020) and the correct equation is:

$$k_{\text{jpl15/20}}(T) = 1.8 \times 10^{-11} \cdot \exp\left(-\frac{600}{T}\right) \quad . \tag{4}$$

We show (Fig. 1) the reaction rate constant in Eq. 4 against the recommendation reported by Sander et al. (2011) and other recent measurements (Leather et al., 2012; Ward and Rowley, 2016).

| $k(298\,\text{K})$ | Source |
|---|---|
| $2.243 \cdot 10^{-12}$ | Sander et al. (2011) |
| $2.404 \cdot 10^{-12}$ | Burkholder et al. (2015, 2020) |
| $2.399 \cdot 10^{-12}$ | Leather et al. (2012) |
| $2.552 \cdot 10^{-12}$ | Ward and Rowley (2016) |

**Table 3.** The rate constant $k$ (in $\text{cm}^3$ molecule$^{-1}$ s$^{-1}$) of reaction R3 ($ClO + CH_3O_2 \longrightarrow$ products) at 298 K. Note that the correct value for the A-factor is: $A = 1.8 \times 10^{-11}$ (which is not listed correctly in recommendations, see eq. 2).

Using the incorrect $A$-factor would result in a rate constant $k_{\text{jpl15/20}}$ which is inconsistent with laboratory measurements. However, when calculating the reaction rate at room temperature $k(298\,\text{K})$, the formulation of Eq. 4 yields consistent results with earlier work and also reproduces the $k(298\,\text{K})$ value reported by Burkholder et al. (2015, 2020), see Table 3. The impact 225 of using the incorrect $A$-factor in model simulations is discussed in section 3.3.2 below.

## 3 Results

### 3.1 Comparing the chemical kinetic and photochemical recommendations by Sander et al., 2011 and Burkholder et al., 2020

A comparison is shown (Fig. 2) for the box model simulation (S1) based on the kinetic and photochemical recommendations 230 by Sander et al. (2011) (which were used in earlier work, Müller et al., 2018; Zafar et al., 2018) and Burkholder et al. (2020), the most recent recommendation (S2). The results for HCl, $ClO_x$, HOCl, $ClONO_2$, and $O_3$ reported earlier (magenta lines in

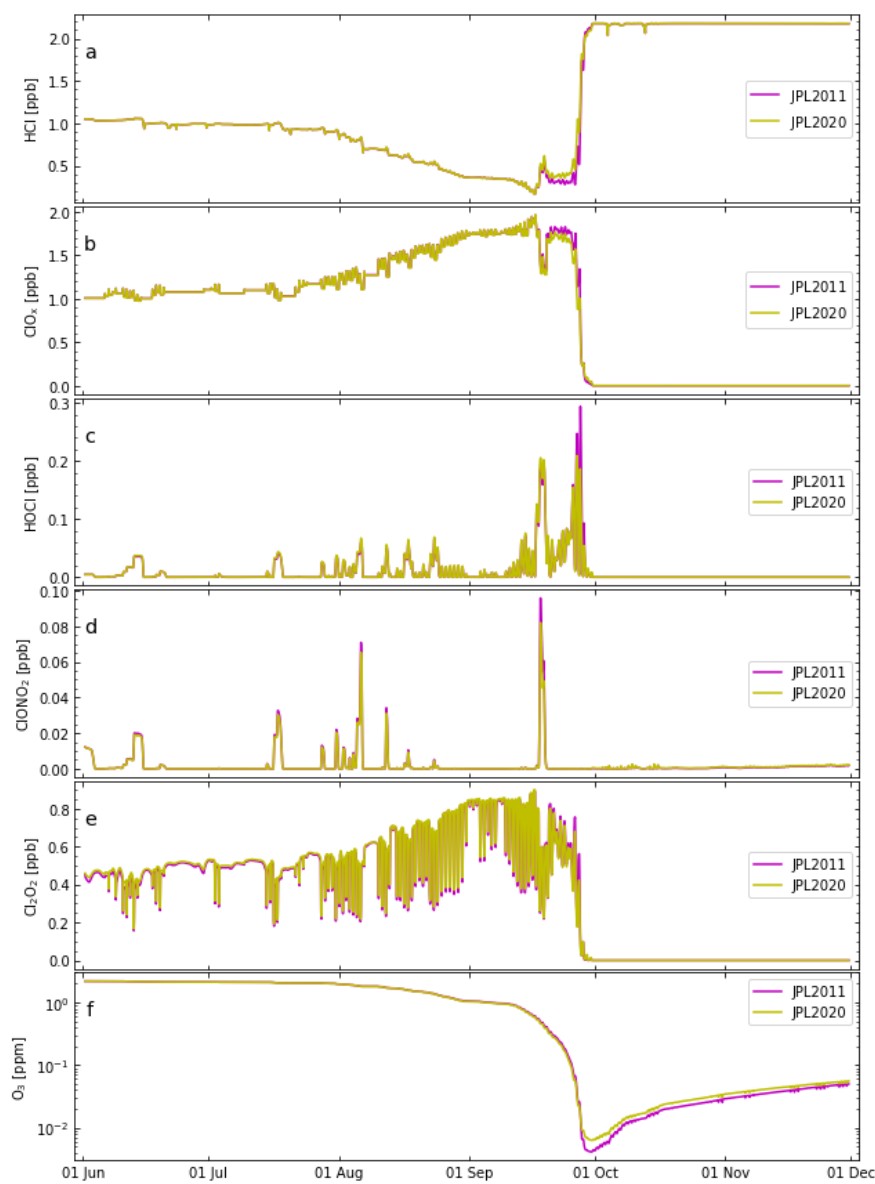

**Figure 2.** Box-model simulations along a trajectory passing through the location of the ozone sonde observation at South Pole of 14 ppb on 74 hPa (391 K) on 24 September 2003 (Grooß et al., 2011; Müller et al., 2018; Zafar et al., 2018, reference trajectory similar as in earlier studies) using the recommendations from Sander et al. (2011) (simulation S1, magenta lines) and Burkholder et al. (2020) (simulation S2, ochre lines). (See also Table 2). Top panel (a) shows HCl (b) $ClO_x$, (c) HOCl, (d) $ClONO_2$, (e) $Cl_2O_2$, and (f) ozone (log-scale). The results for simulation S1 using Sander et al. (2011) are identical to those reported earlier (Müller et al., 2018; Zafar et al., 2018).

Fig. 2; simulation S1 in Table 2) showed that at 16–18 km (85–55 hPa) in the core of the vortex, high levels of active chlorine are maintained by HCl null cycles, where the formation of HCl is balanced by immediate reactivation (Müller et al., 2018; Zafar et al., 2018). The strongest ozone loss rates occur in September. The results when using the most recent recommendation (Burkholder et al., 2020) are very similar to those reported earlier (Fig. 2). However, levels of HCl in September are somewhat lower (and thus $ClO_x$ somewhat higher, resulting in somewhat stronger ozone loss) when using the recommendations by Sander et al. (2011). The maximum difference in ozone between the two runs using different kinetic recommendations (Fig. 2) is less than 0.025 ppm (or 25 ppb). Overall, the differences between simulations S1 and S2 are very small (Fig. 2); see also section 2.3.1.

## 3.2  The impact of initial water vapour

### 3.2.1  Formation of ice particles

Two simulations are compared for a different initialisation of $H_2O$, namely $H_2O = 4.11$ ppm (simulation S2) and $H_2O = 2.05$ ppm (a more realistic initialisation for $H_2O$, simulation S3); both simulations are employing the most recent kinetic recommendations (Burkholder et al., 2020). For an initial water vapour molar mixing ratio of 2.05 ppm, both the area of ice

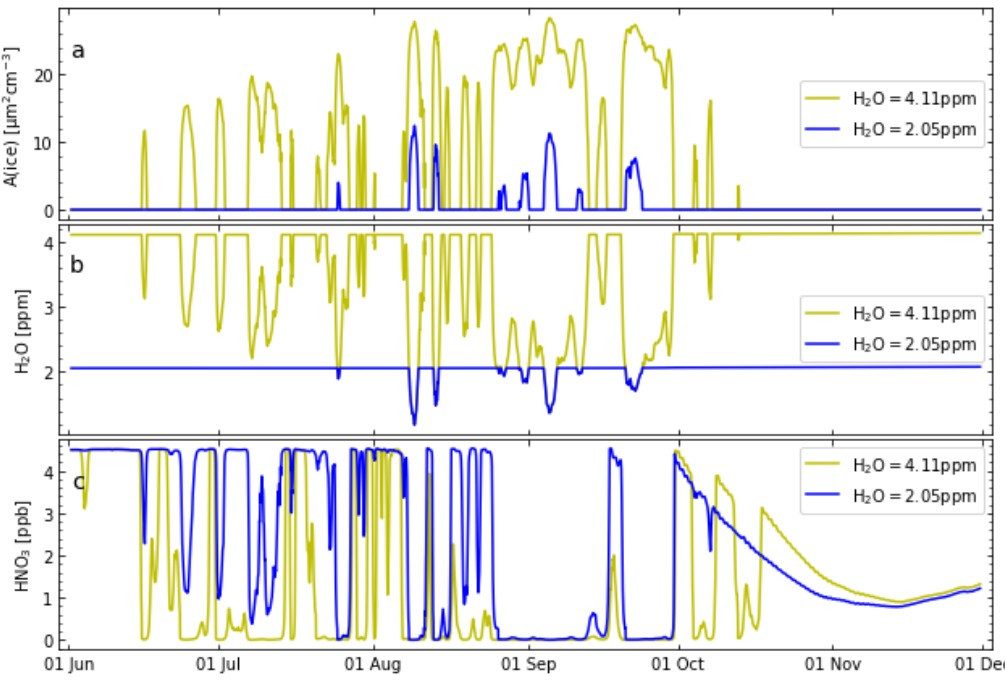

**Figure 3.** The simulated temporal development of ice surface (top panel, a), gas-phase water vapour mixing ratios (middle panel, b) and gas-phase $HNO_3$ mixing ratios (bottom panel, c). The results for both initial water vapour molar mixing ratio of 4.11 ppm (simulation S2, ochre lines) and 2.05 ppm (simulation S3, blue lines) are shown. (See also Table 2 for further details).

PSCs and the molar mixing ratio of gas-phase water vapour are substantially smaller than for an initial water vapour molar mixing ratio of 4.11 ppm (Fig. 3). In particular, the simulated ice surface for $H_2O_{ini} = 2.05$ ppm is substantially smaller than for $H_2O_{ini} = 4.11$ ppm. There should be observational consequences of the very different ice surfaces in simulations S2 and S3 (Fig. 3), i.e. observations should allow discriminating between the hypotheses about initial water vapour in simulations S2 and S3. A significant difference between the runs for a different initial water vapour molar mixing ratio is a different concentration

of gas-phase $HNO_3$ – caused by the uptake of $HNO_3$ onto ice particles (Fig. 3, bottom). The enhanced gas-phase $HNO_3$ (for initial water vapour molar mixing ratio of 2.05 ppm) allows more $NO_x$ to be released to the gas phase and (in sun-light) leads to somewhat more formation of $ClONO_2$ (Fig. 4).

However, the very different ice surfaces between simulations S2 and S3 have remarkably little impact on the temporal development of HCl and active chlorine (although there are a few periods with more $ClONO_2$ for an initial water vapour

molar mixing ratio of 4.11 ppm, Fig. 4). This has consequences for chemical ozone depletion (Fig. 4). There is a slightly lower minimum value of ozone ($\approx 10$ ppb lower) for an initial water vapour molar mixing ratio of 4.11 ppm. The simulated ozone values for an initial water vapour molar mixing ratio of 2.05 ppm are very similar to those for an initial water vapour molar mixing ratio of 4.11 ppm in June and July and are $\approx 50$ ppb higher between August and mid-September; the largest *difference* ($\approx 100$ ppb) in ozone molar mixing ratios is reached in mid to late September.

This finding is consistent with the notion that the *rate constant* of the heterogeneous reactions within HCl null cycles is of little relevance for the efficacy of the HCl null cycles (Müller et al., 2018). Although it is important for the efficacy of the HCl null cycles that temperatures are sufficiently low so that particles are present and heterogeneous reactions occur. The rate constant of heterogeneous reactions is influenced strongly by the type of the available PSC particles. The efficacy of the HCl null cycles, however, is limited by the rates of the reactions of Cl with $CH_4$ and $CH_2O$ (see appendix A). In consequence,

a substantial difference in initial water vapour molar mixing ratios in simulations S2 and S3 does not result in a substantial difference of polar chlorine chemistry and ozone loss (Fig. 4).

### 3.2.2 The eruption of the Hunga volcano

The initial water vapour in the Antarctic vortex assumed here and for the related model simulations (Table 1 and section 3.2.1) is discussed below regarding the interpretation of water vapour injections into the stratosphere by volcanic eruptions. In January

2022, the eruption of the Hunga underwater volcano injected a huge amount of water vapour, unprecedented in the observational record, into the mid-stratosphere (Wohltmann et al., 2023; Fleming et al., 2024; Zhou et al., 2024).

The impact of this water vapour enhancement on Antarctic ozone has been assessed through model studies. Fleming et al. (2024) find that the excess $H_2O$ is projected to increase polar stratospheric clouds and springtime halogen-ozone loss, enhancing the Antarctic ozone hole by 25–30 DU. Wohltmann et al. (2023) find that the direct chemical effect of the increased water

vapour on vortex average Antarctic ozone depletion in June through October was minor (less than 4 DU). Zhou et al. (2024) confirm this conclusion, but find somewhat more ozone loss caused by the injected water vapour ($\approx 10$ DU) at the vortex edge.

The impact of the stratospheric water vapour enhancement through the Hunga eruption on Antarctic ozone has further been assessed in the analysis of satellite observations (Santee et al., 2024). It was observed that the Hunga eruption increased the

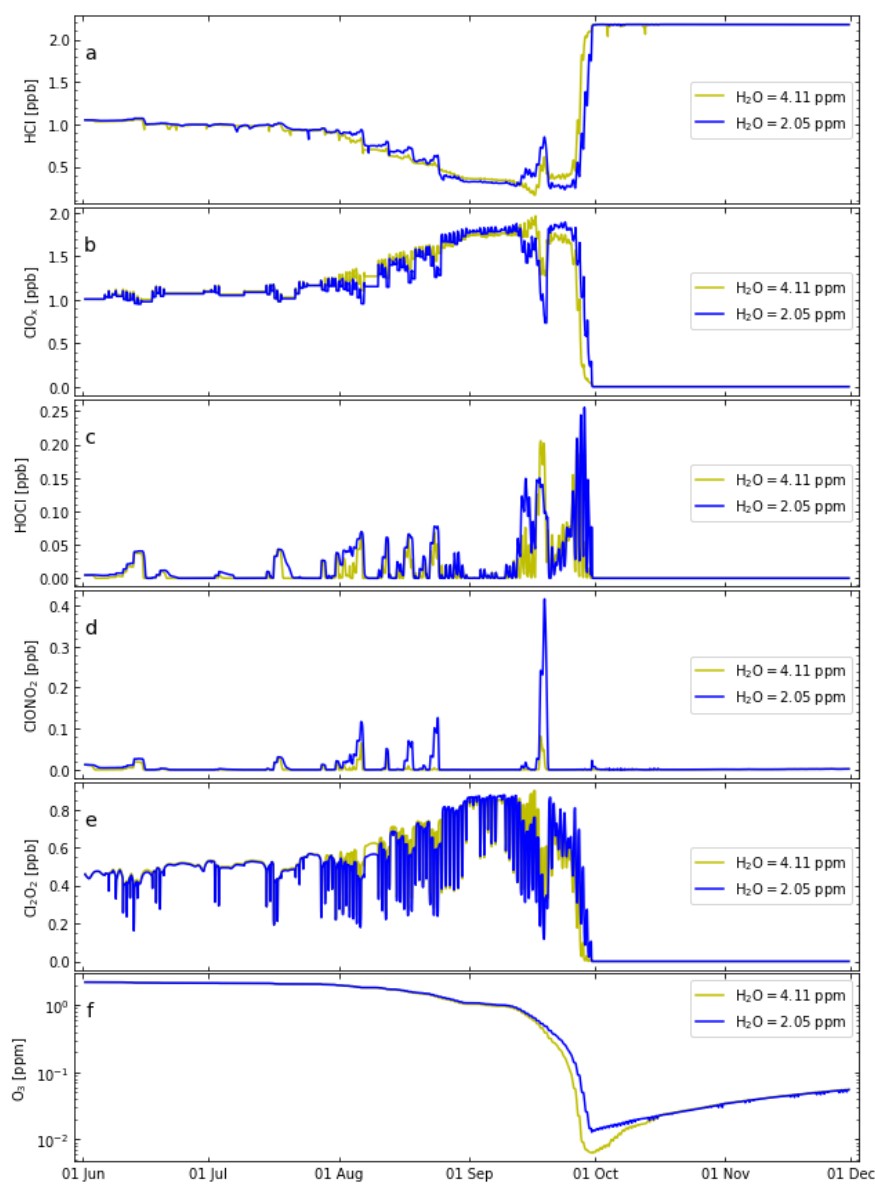

**Figure 4.** Similar as Fig. 2 but using the recommendations from Burkholder et al. (2020). Results are compared for a different initialisation of $H_2O$; $H_2O = 4.11$ ppm (simulation S2, ochre lines) and $H_2O = 2.05$ ppm (simulation S3, blue lines). (See also Table 2).

vertical extent of PSC formation and chlorine activation in early Austral winter in the Antarctic vortex in 2023 (the Antarctic season influenced most strongly by the Hunga eruption). Nonetheless, ozone depletion in the Antarctic in 2023 was unremarkable throughout the lower stratosphere (Santee et al., 2024). The observation of a small impact of water vapour injected into the stratosphere on polar ozone loss is consistent with the notion put forward in this paper (Fig. 3) that low temperatures in the vortex, which occur regularly in the Antarctic, limit the atmospheric water vapour to the water vapour saturation pressure over ice and thus remove any anomalies through dehydration before they can affect ozone loss.

The minor impact of the huge water vapour injections into the stratosphere by the Hunga volcano on Antarctic ozone in the 2023 season (Wohltmann et al., 2023; Fleming et al., 2024; Zhou et al., 2024; Santee et al., 2024) is consistent with the small impact of initial water vapour in mid-winter and the subsequent formation of ice PSC particles in the model simulations presented here (section 3.2.1 and Fig. 3). First, the low temperatures in the lower stratosphere in the core of the Antarctic vortex determine mid-winter water vapour (independent of the amount of water vapour present at the time of the formation of the vortex). Second, even if higher water vapour mixing ratios prevailed in mid-winter, chlorine activation and chemical ozone loss remain practically unaltered (Fig. 4).

## 3.3   The impact of initial HCl

### 3.3.1   Reference simulation

We conducted a simulation (for $H_2O_{initial} = 2.05$ ppm) assuming $HCl_{initial} = 0$ (see Sec. 2.2), which corresponds to an increase in initial active chlorine ($ClO_{x,initial} = 2.26$ ppb); that is we assume $Cl_y$ = const. (Fig. 5). Assuming an initial value of HCl = 0 for early June (red lines in Fig. 5) resembles the conditions in the atmosphere (Wohltmann et al., 2017; Grooß et al., 2018, see also section 2.2 above), while an initial value of HCl = 1.05 ppb (blue lines in Fig. 5) is closer to HCl values in current model simulations. Therefore, we refer to simulation S4 here as the reference simulation.

As expected, for more initial $ClO_x$, there is a somewhat stronger ozone depletion (Fig. 5, bottom panel, red line). The difference in ozone between the two simulations is below $\approx 100$ ppb in June and July but increases to $\approx 400$ ppb in September. However, the ozone minimum values reached, differ only by $\approx 10$ ppb. Overall, the difference in absolute ozone depletion between the two simulations (S3 and S4) is moderate (albeit not in relative terms) in accordance with the conclusions by Grooß et al. (2018).

However, there is clearly an earlier onset of strong ozone depletion when $HCl_{initial} = 0$ is employed, with the difference between simulation S3 and S4 notable in late August/early September. Although these different temporal developments of ozone (and $ClO_x$) are obvious in our Lagrangian simulation, it will not be simple to detect such a behaviour in satellite observations, where spatial averages over large horizontal and vertical scales are measured for a given point in time.

Further, HOCl molar mixing ratios are substantially higher for $HCl_{initial} = 0$ (in particular from mid-June to mid-August). The reduction of HOCl via the heterogeneous reaction R2 is suppressed for low HCl (Fig. 5). Indeed, the HCl molar mixing ratios remain low from June to mid September indicating that the HCl null cycles are effective in maintaining low HCl molar

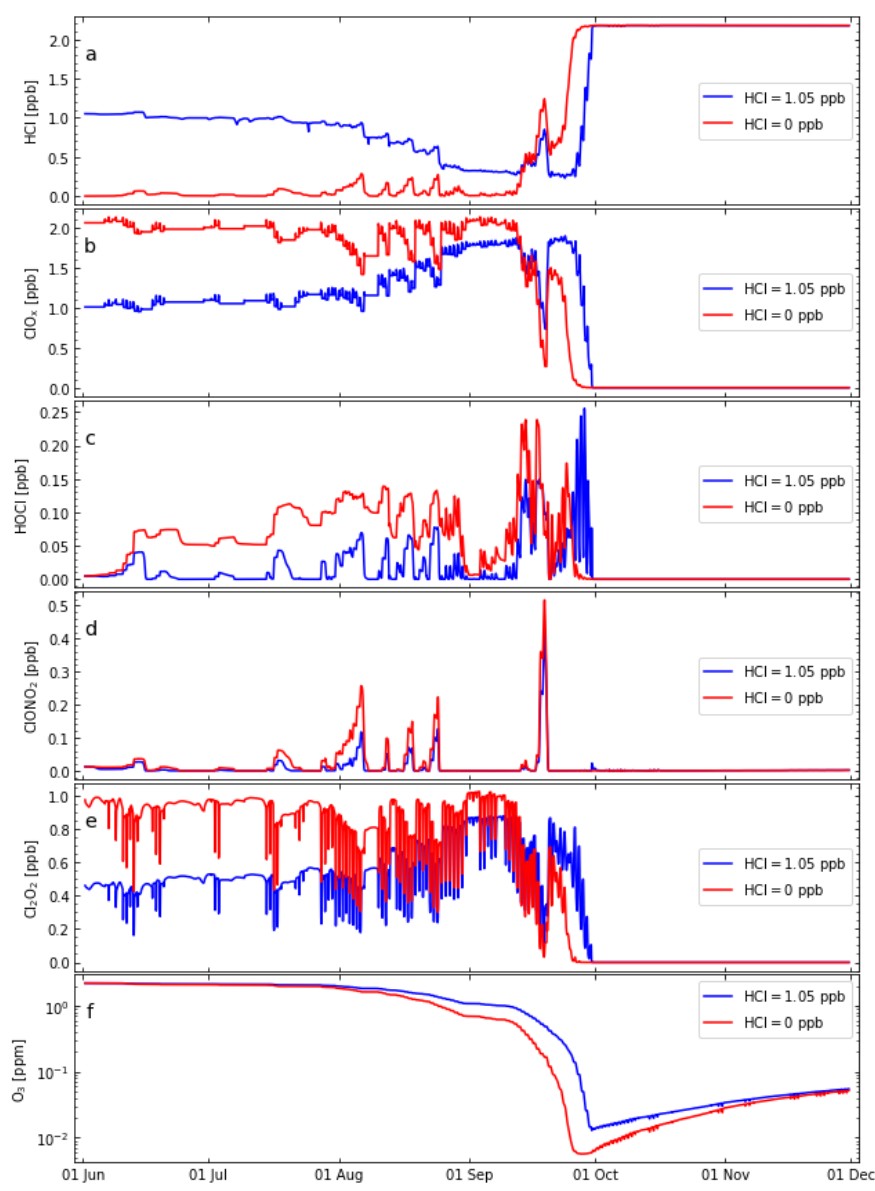

**Figure 5.** Similar as Fig. 4 (for initial $H_2O = 2.05$ ppm) but comparing the results for simulation S3 (blue lines) with the results for simulation S4 (red lines), where a different initialisation of HCl and active chlorine ($ClO_x$) was used. The kinetic recommendations of Burkholder et al. (2020) are used. (See also Table 2).

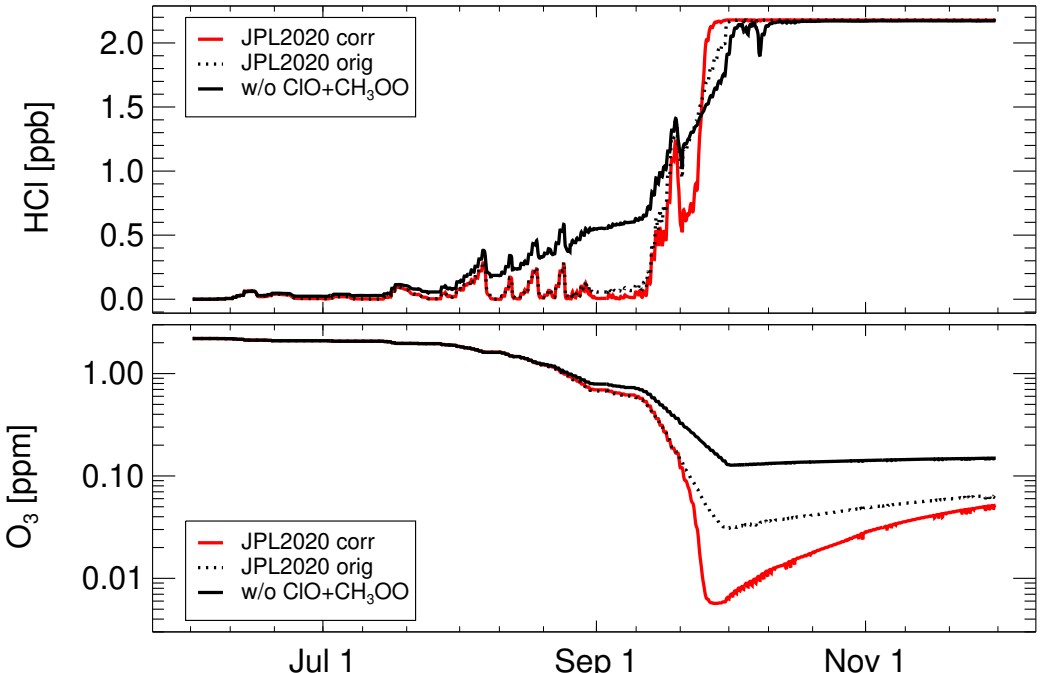

**Figure 6.** The impact of the formulation of reaction $CH_3O_2 + ClO$ (see section 2.3.2) in simulation S4. Red line shows the results for simulation S4 (as in Fig. 5), dotted black line the results assuming the incorrect $A$-factor (see section 2.3.2) and the solid black line neglecting reaction $CH_3O_2 + ClO$ (R3) entirely. Top panel shows molar mixing ratios for HCl, bottom panel for ozone.

mixing ratios (Müller et al., 2018). When ozone molar mixing ratios reach extremely low values in late September, HCl molar mixing ratios increase rapidly, which occurs a few days earlier in the case of $HCl_{initial} = 0$ (Fig. 5).

Under the conditions discussed here, values of $ClONO_2$ remain strongly depressed (close to zero, with few exceptions; Fig. 5 panel d). This statement is true for the entire simulated period, including the period of strongest ozone loss throughout September. This observation is consistent with the dominance of the HCl null cycles that do not involve the heterogeneous reaction R1 (Müller et al., 2018, see also appendix A). Differences in the temporal development of HOCl, $ClO_x$, and HCl between simulations S3 and S4 are greater than for ozone and thus might possibly be simpler to detect in satellite observation than a different temporal development of ozone.

### 3.3.2 The impact of the reaction $CH_3O_2 + ClO$

Reaction R3 is essential for the HCl null cycle initiated by the reaction of Cl with $CH_4$ (C1, see AR1-AR8 in appendix A) and thus has a certain importance for "ozone hole" chemistry (e.g. Crutzen et al., 1992; Zafar et al., 2018). Here we report further results of simulation S4 exploring the impact of assuming the incorrect $A$-factor in the rate constant of reaction R3 (see section 2.3.2 above). Clearly, the incorrect $A$-factor leads to underestimating the rate constant for reaction R3 by a factor of

ten. Assuming this incorrect value (for R3) in simulation S4 yields a slower HCl increase in late September and the minimum ozone molar mixing ratio reached is 30.6 ppb instead of 5.7 ppb in simulation S4 (Fig. 6).

Of course, assuming a too low (by a factor of ten) value for the rate constant of reaction R3 is not equivalent to neglecting reaction R3 completely. Removing reaction R3 completely from the set of reactions considered here leads to a weaker rate of ozone depletion. Under these conditions, the minimum ozone is more than 100 ppb, whereas the simulated minimum ozone is 5.7 ppb, when reaction R3 is taken into account correctly (Fig. 6). Further, the temporal development of HCl is very different; when neglecting reaction R3, the increase in HCl starts earlier and is much less steep (Fig. 6). Using the value for the rate constant of reaction R3 as listed in the recommendations (Burkholder et al., 2015, 2020) shows very low HCl values on 10 September (below 0.1 ppb), whereas HCl on this day is about 0.6 ppb when reaction R3 is neglected. Using the correct value for reaction R3 (Eq. 4) results in even lower HCl values (close to zero).

### 3.4 Multi-trajectory simulations

In the discussions above, one particular (reference) trajectory was considered. However, this trajectory is representative for the conditions in the core of the Antarctic vortex at 16–18 km (85–55 hPa, 390–430 K). To demonstrate this, we selected twenty one trajectories passing the South Pole (in late September/early October) at the 400 K potential temperature level; these trajectories include diabatic descent and latitude variations. This selection is the same as in earlier work (Grooß et al., 2011; Müller et al., 2018). For the simulations we use the most recent kinetic recommendations (Burkholder et al., 2020, with the corrected rate constant of R3). From early August to early October these trajectories show roughly the same diabatic descent of $\approx 10\,\mathrm{K}$, similarly as the reference trajectory discussed above. However, over this period, the different trajectories show strong variations in latitude (and thus exposure to sunlight); the latitude varies between the South Pole and $\approx 65°\mathrm{S}$ with some equatorward excursions to $\approx 60°\mathrm{S}$ and, sometimes, to $\approx 55°\mathrm{S}$.

The initial values (for 1 August) were taken from Grooß et al. (2011); denitrification and dehydration are taken into account, total chlorine ($Cl_y$) is deduced from observations (correlation with $N_2O$) and the HCl initialisation is taken from a climatology based on ACE-FTS (Atmospheric Chemistry Experiment – Fourier Transform Spectrometer) measurements (Jones et al., 2012). Overall, these initial conditions correspond well with the assumptions made for simulation S4. Since the initial value of $Cl_y$ is taken from Grooß et al. (2011), also the initial $ClO_x$ is different for the individual trajectories.

The results of the multi-trajectory simulations (Fig. 7) show a certain variability in the ozone loss rate, which depends strongly on solar insolation and on the initial value of $ClO_x$; also note that initial ozone is different for the individual trajectories. The minimum ozone is reached for the individual trajectories between late September and early October. There is also a certain variability of the temporal development of HCl, with some trajectories showing intermittent increases in HCl for certain periods (but no complete deactivation).

Nonetheless, all trajectories show strongly enhanced values of $ClO_x$ over the period of strong ozone loss in August and September, consistent with suppressed values of HCl (Fig. 7). Minimum ozone values for all trajectories are very low; below $\approx 100$ ppb for most and below 10 ppb for several trajectories (Fig. 7). As in the reference simulation, the period of rapid ozone loss (driven by high levels of $ClO_x$) ends abruptly with chlorine deactivation through very rapid formation of HCl (e.g.,

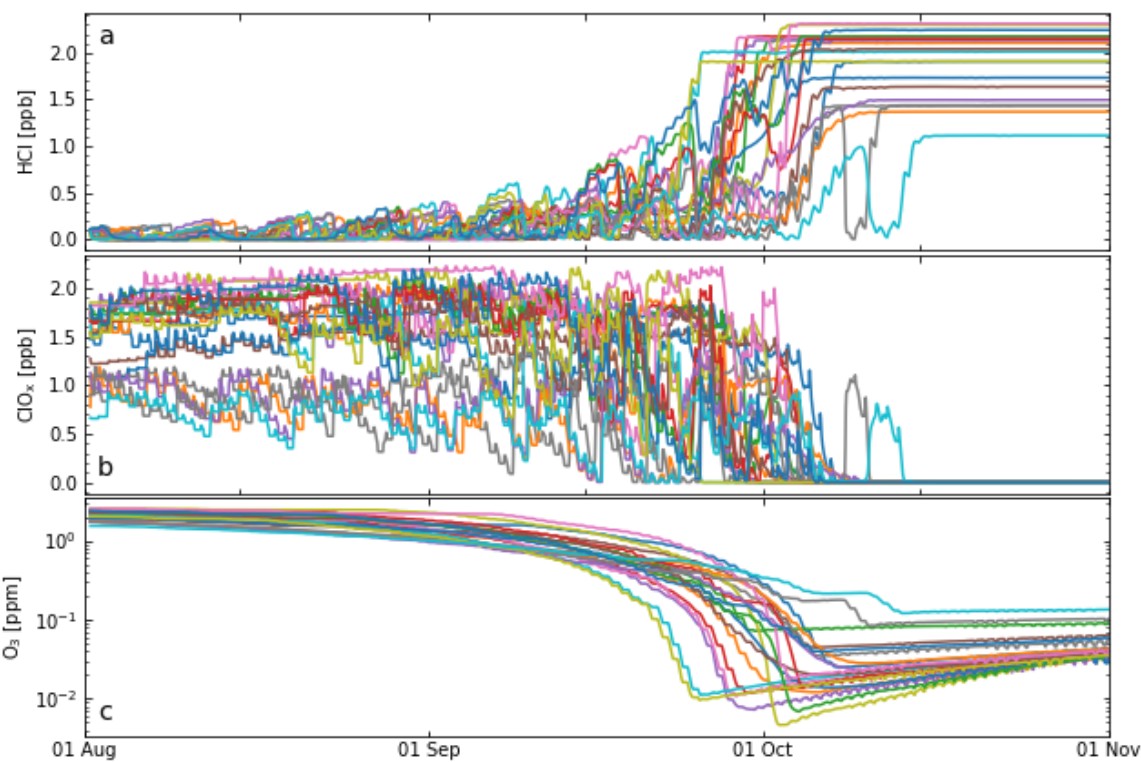

**Figure 7.** Results from multi-trajectory simulations (21 trajectories) of the CLaMS box-model. The trajectories are consistently initialised (for 1 August) as in Grooß et al. (2011), which corresponds well with the assumptions made for simulation S4 (see text for details). Simulations were performed for a set of trajectories passing the South Pole at 400 K (Grooß et al., 2011). The results of the box-model simulations are shown for the time period from 1 August to 1 November 2003. Panel (a) shows HCl, panel (b) $ClO_x$, and panel (c) ozone. Individual trajectories are shown in different colour to allow them to be distinguished more easily.

Crutzen et al., 1992; Douglass et al., 1995; Grooß et al., 1997; Müller et al., 2018). After deactivation, HCl values remain high and practically unchanged in the box model simulation.

## 4   Discussion

Simulations of Antarctic chlorine and ozone chemistry for the core of the Antarctic vortex (16–18 km, 85–55 hPa, 390–430 K) indicate that HCl null cycles (C1 and C2, see appendix A) are effective throughout winter and spring (Grooß et al., 2011; Müller et al., 2018). The HCl null cycles require sufficiently low temperatures so that heterogeneous reactions (in particular reaction R2) have significant reaction rates. Further, a significant rate of the reaction $CH_3O_2 + ClO$ (reaction R3) is important for the efficacy of the HCl null cycle C1 (as discussed earlier, Crutzen et al., 1992; Zafar et al., 2018, see also appendix A).

The HCl null cycles allow HCl molar mixing ratios to be maintained at very low values so that rapid ozone depletion proceeds until the deactivation of $ClO_x$ into HCl.

A low value of water vapour in mid winter is suggested here ($H_2O_{ini} = 2.05$ ppm, see also section 2.2.1) to account for the observed dehydration in the Antarctic vortex (e.g., Kelly et al., 1989; Vömel et al., 1995; Nedoluha et al., 2002). This leads to substantially less formation of ice particles (Fig. 3) and thus to a substantially lower rate of heterogeneous reactions on ice as well as less uptake of $HNO_3$ from the gas phase compared to earlier work (Müller et al., 2018; Zafar et al., 2018). However, ozone depletion is not strongly affected, consistent with the results of Kirner et al. (2015), who used the chemistry-climate model ECHAM5/MESSy for Atmospheric Chemistry (EMAC) to investigate the impact of different types of PSCs on Antarctic chlorine activation and ozone loss. Kirner et al. (2015) find that heterogeneous chemistry on liquid particles is responsible for more than 90% of the ozone depletion in Antarctic spring and that heterogeneous chemistry on ice particles causes less than 5 DU of additional column ozone depletion. The conclusion that heterogeneous chlorine chemistry is dominated by reactions on liquid particles is supported also by other work (e.g., Wegner et al., 2012; Grooß and Müller, 2021; Tritscher et al., 2021).

Further, Antarctic winter MLS observations during the period of the onset of chlorine activation (between May and July in austral winter) and ground-based measurements at Syowa station in early July (Nakajima et al., 2020) indicate that very low molar mixing ratios of HCl prevail in the vortex, which are not well reproduced by model simulations (Wohltmann et al., 2017; Grooß et al., 2018). Possible reasons for these very low molar mixing rations of HCl are discussed in detail above (section 2.2.2).

Accounting for very low HCl values in mid-winter in the initial values of our simulations (see also section 2.2.2), we find very low HCl molar mixing ratios throughout winter and spring. This result is consistent with HCl values being maintained at very low values through the efficacy of the HCl null cycles (C1 and C2, see appendix A). Further, ozone depletion is affected by the initial values of HCl, namely the timing of maximum ozone loss. However, the minimum values of Antarctic ozone reached are similar, consistent with Grooß et al. (2018).

## 5    Conclusions

The results of our simulations corroborate earlier findings that effective HCl null cycles (C1 and C2; see Appendix A) allow high levels of active chlorine to be maintained in the Antarctic lower stratosphere during the period of strong ozone depletion. During this period, HCl production rates in the gas-phase are high (and increase with decreasing ozone, Grooß et al., 2011; Müller et al., 2018).

The sensitivity investigations in the present study show the following. First, using the most recent recommendation for chemical kinetic and photochemical data (Burkholder et al., 2020), does not change the results of the simulations substantially compared to earlier work (where Sander et al., 2011, was used). Second, the HCl null cycles require the heterogeneous reaction R2 ($HCl + HOCl \rightarrow Cl_2 + H_2O$) to proceed; i.e., temperatures need to be sufficiently low. Further, the gas-phase reaction R3 ($ClO + CH_3O_2 \rightarrow$ prod.) is essential (for the null cycle initiated by the reaction $CH_4 + Cl$, see also appendix A and sec-

tion 2.3.2). If reaction R3 was neglected, HCl molar mixing ratios in early September of $\approx 0.6$ ppb are simulated, instead of HCl values close to zero. Further simulated minimum ozone is more than 100 ppb instead of $\approx 6$ ppb with reaction R3.

400    Third, taking into account the observed dehydration in the Antarctic lower stratosphere in winter (see section 2.2), which was not properly accounted for in earlier work (Müller et al., 2018; Zafar et al., 2018), substantially reduces the occurrence of ice clouds in the model, but does not affect strongly the results of chlorine chemistry and ozone loss. The most important impact of the simulated difference in the occurrence of ice clouds in the model is caused by the uptake of $HNO_3$ from the gas-phase into ice particles. The $HNO_3$ uptake is smaller, when less ice particle surface is available. If the observed dehydration is taken

405    into account in the simulations, a slightly higher minimum value of ozone ($\approx 10$ ppb higher) is simulated.

Finally, the maximum impact of the observed (but unexplained) observation of extremely low values of HCl during polar night (Wohltmann et al., 2017; Grooß et al., 2018) was investigated based on the current simulations. This is done by assuming an HCl molar mixing ratio of zero after polar night on 1 June (while keeping $Cl_y$ constant). These assumptions lead to a temporal development of the chlorine chemistry that is different from that assuming a higher initial HCl; in particular HOCl

410    molar mixing ratios are enhanced from about mid-June to mid-August. Further, $ClO_x$ is enhanced throughout winter and spring, and HCl molar mixing ratios remain very low until rapid chlorine deactivation occurs into HCl. Also the strength of the ozone loss rate and the timing of maximum ozone loss is affected by the initial value of HCl, but not the minimum ozone value (consistent with Grooß et al., 2018); the simulated ozone minimum values differ by $\approx 10$ ppb. Overall, our simulations indicate extremely low minimum ozone values at the South Pole (below 50 ppb) in late September/early October in agreement

415    with observations (Solomon et al., 2005; Grooß et al., 2011; Johnson et al., 2023).

*Code availability.*   The CLaMS model is accessible via https://jugit.fz-juelich.de/clams/clams-git.git

*Data availability.*   The model results presented here are attached to the paper as an electronic supplement (in netcdf format).

## Appendix A: HCl null cycles

The HCl null cycles listed below are responsible for the maintenance of high levels of active chlorine throughout Antarctic spring and were reported earlier (Müller et al., 2018; Zafar et al., 2018); they are repeated here for reference. Cycle C1 (Crutzen et al., 1992; Müller et al., 2018) starts with HCl production in the reaction $CH_4 + Cl$

$$CH_4 + Cl \rightarrow HCl + CH_3 \tag{AR1}$$

$$CH_3 + O_2 + M \rightarrow CH_3O_2 + M \tag{AR2}$$

$$CH_3O_2 + ClO \rightarrow CH_3O + Cl + O_2 \tag{AR3}$$

$$CH_3O + O_2 \rightarrow HO_2 + CH_2O \tag{AR4}$$

$$ClO + HO_2 \rightarrow HOCl + O_2 \tag{AR5}$$

$$HOCl + HCl \rightarrow Cl_2 + H_2O \tag{AR6}$$

$$Cl_2 + h\nu \rightarrow 2\,Cl \tag{AR7}$$

$$Cl + O_3 \rightarrow ClO + O_2 \quad (2\times) \tag{AR8}$$

$$\text{Net(C1)}: \quad CH_4 + 2\,O_3 \rightarrow CH_2O + H_2O + 2\,O_2$$

Further, the formation of HCl in the reaction

$$CH_2O + Cl \rightarrow HCl + CHO \tag{AR9}$$

leads to the following cycle (C2, Müller et al., 2018):

$$CH_2O + Cl \rightarrow HCl + CHO \tag{AR10}$$

$$CHO + O_2 \rightarrow CO + HO_2 \tag{AR11}$$

$$ClO + HO_2 \rightarrow HOCl + O_2 \tag{AR12}$$

$$HOCl + HCl \rightarrow Cl_2 + H_2O \tag{AR13}$$

$$Cl_2 + h\nu \rightarrow 2\,Cl \tag{AR14}$$

$$Cl + O_3 \rightarrow ClO + O_2 \tag{AR15}$$

$$\text{Net(C2)}: \quad CH_2O + O_3 \rightarrow CO + H_2O + O_2$$

## Appendix B: Changes in JPL2015 and JPL2020

In Burkholder et al. (2020) a new algorithm for the formulation of termolecular reactions (association dissociation) was introduced, which is used here. Note the correction (compared to the values listed in the recommendations) to the reaction $ClO + CH_3O_2 \rightarrow$ prod. discussed in detail in section 2.3.

| | | |
|---|---|---|
| $Cl + CH_4$ | $\rightarrow$ | $HCl + CH_3$ |
| $ClO + CH_3O_2$ | $\rightarrow$ | prod. |
| $CFC12 + O(^1D)$ | $\rightarrow$ | prod. |
| $HCFC22 + O(^1D)$ | $\rightarrow$ | prod. |
| $CFC113 + O(^1D)$ | $\rightarrow$ | prod. |
| $HCFC22 + OH$ | $\rightarrow$ | prod. |
| $CH_3Cl + OH$ | $\rightarrow$ | prod. |
| $HO_2 + NO_2$ | $\rightarrow$ | $HO_2NO_2$ |
| $HO_2NO_2$ | $\rightarrow$ | $HO_2 + NO_2$ |
| $NO_2 + NO_3$ | $\rightarrow$ | $N_2O_5$ |
| $ClO + ClO$ | $\rightarrow$ | $Cl_2O_2$ |
| $Cl_2O_2$ | $\rightarrow$ | $ClO + ClO$ |
| $BrO + NO_2$ | $\rightarrow$ | $BrONO_2$ |
| $OH + CO$ | $\rightarrow$ | $CO_2 + H$ |
| $CH_2O + h\nu$ | $\rightarrow$ | prod. |

**Table A1.** Changes between recent recommendations; Sander et al. (2011) versus Burkholder et al. (2015).

| | | |
|---|---|---|
| $CO + OH$ | $\rightarrow$ | $CO_2 + H$ |
| $O(^3P) + NO_2$ | $\rightarrow$ | $NO + O_2$ |
| $HO_2 + HO_2$ | $\rightarrow$ | $H_2O_2 + O_2$ |
| $HNO_3 + OH$ | $\rightarrow$ | $NO_3 + H_2O$ |
| $HO_2 + NO$ | $\rightarrow$ | $NO_2 + OH$ |
| $NO + NO_3$ | $\rightarrow$ | $NO_2 + NO_2$ |
| $H + O_2$ | $\rightarrow$ | $HO_2$ |
| $CH_3O_2NO_2 + h\nu$ | $\rightarrow$ | $CH_3O_2 + NO_2$ |

**Table A2.** Changes between recent recommendations; Burkholder et al. (2015) versus Burkholder et al. (2020).

425 *Author contributions.* Y.Z.-L., J.-U.G., and A.M.Z. conducted the simulations with the CLaMS box model that are reported here. R.M. and J.-U.G. conceived and designed the research project. The issues regarding the rate constant of the reaction ClO + $CH_3O_2$ (as discussed in section 2.3) were first raised by R.L.. All co-authors discussed the results and contributed to formulating the manuscript.

*Competing interests.* J.-U.G. and R.M. are editors of ACP; otherwise the authors declare that they have no competing interests

*Acknowledgements.* This paper originates from the Master's thesis of Y. Z-L.; we thank Prof. E. Rühl from the Free University of Berlin for supervising this thesis work. We also thank J. Burkholder for very helpful comments on the rate constant of reaction R3. We are grateful to M. Chipperfield and D. Kinnison for comments on the manuscript. We thank the European Centre for Medium-range Weather Forecasts (ECMWF) for providing meteorological data sets. This research was partly funded by the "Pilot Lab Exascale Earth System Modelling" project of the Helmholtz association. We thank the three anonymous reviewers of the paper very much for helpful comments that led to an improved version of the paper.

430

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
