# Peer review of "The impact of dehydration and extremely low HCl values in the Antarctic stratospheric vortex in mid-winter on ozone loss in spring"

_EGUsphere, 2024_

## Author Comment (AC1)

*Reply to comments by reviewer one on*

**The impact of dehydration and extremely low HCl values in the Antarctic stratospheric vortex in mid-winter on ozone loss in spring**

*by Yiran Zhang-Liu et al.*

We thank the reviewer very much for her/his interest in our paper and for very helpful comments. The comments are repeated below in blue and a point-by-point response is given in normal font and black colour.

The paper has been revised in view of the comments by the reviewer.

**Reviewer one**

Summary: The paper addresses three questions: What is the impact of updates to previous recommendations on chemical kinetics on Antarctic ozone depletion? Furthermore, while dehydration strongly regulates Antarctic stratospheric water vapour, its impact on ozone depletion is small. And thirdly, an HCl null cycle and a further cycle starting with $CH_2O + Cl \longrightarrow HCl + CHO$ contribute substantially to keeping HCl low and $ClO_x$ high, hence leading to enhanced ozone depletion.

I learnt a few things reading the paper. I had not thought about the two null cycles and their role in sustaining ozone depletion. The prevailing view is that $CH_4 + Cl$ is a termination reaction for ozone depletion, not the start of yet another cycle of ozone depletion and a null cycle for HCl. Also the typo / order-of-magnitude error in the reaction $ClO + CH_3O_3$ is good to know about – that might be wrong in many chemistry models. The paper represents good, solid work, enhancing our understanding of chemical kinetics of the Antarctic polar vortex. Of course this topic is sometimes considered to be fairly mature, but this paper presents a fresh take on this subject. I don't have many comments to make; the method is fairly straightforward. It involves trajectory calculations simulating atmospheric chemistry under Antarctic conditions and testing the sensitivity of the results to assumptions on initial values for HCl and water, and for correcting the typo in the rate coefficient.

I recommend publication of the paper in ACP subject to addressing the small, technical comments below.

Thank you very much for your comments on our paper. All your comments have been taken into account when producing a revised version of the paper.

**Comments:**

Table 1: Here and throughout the text, I suggest to put "volume" in front of "mixing ratio", and to use units of ppmv, ppbv, etc, instead of ppm and ppb. Otherwise these can be misunderstood.

We certainly agree with the reviewer that a confusion of volume and mass mixing ratios should be avoided. Thus, the manuscript must be changed. However, our concept in the paper is that molar mixing ratios are shown; molar mixing ratios are identical to volume mixing ratios for an ideal gas. And the deviation of most gases discussed in our manuscript from an ideal gas behaviour can hardly be measured. We have changed the manuscript. In Table 1, we now say "molar mixing ratio" in the caption, and, more importantly perhaps, we now say "molar mixing ratios" throughout the manuscript. We have also inserted the following explanation "(Molar mixing ratios are identical to volume mixing ratios in the case of an ideal gas)" into the introduction of the paper now.

Section 3.2: Can a line be drawn from the small impact of the initial value of $H_2O$ on chlorine and ozone to the (thus far) small impact of the increased water vapour in the stratosphere since the Hunga-Tonga Hunga-Haapai eruption? There had been some expectation in the community that this would increase ozone depletion, but the 2023 season was quite ordinary.

We agree with this comment and we have extended Sec. 3.2. The reviewer is correct in pointing out the relevance of our results in section 3.2. to the Hunga eruption. Indeed, the impact on Antarctic ozone is small as implied by the reviewer comment.

In response to the comment we have added the following discussion to the manuscript in section 3.2:

"The initial water vapour in the Antarctic vortex assumed here and the related model results (...) are discussed below regarding the interpretation of water vapour injections into the stratosphere by volcanic eruptions. In

January 2022, the eruption of the Hunga underwater volcano injected a huge, unprecedented in the observational record, amount of water vapour into the mid-stratosphere (Wohltmann et al., 2023; Fleming et al., 2024; Zhou et al., 2024).

The impact of this water vapour enhancement on Antarctic ozone has been assessed through model studies. (Fleming et al., 2024) find that the excess $H_2O$ is projected to increase polar stratospheric clouds and spring-time halogen-ozone loss, enhancing the Antarctic ozone hole by 25–30 DU. Wohltmann et al. (2023) find that the direct chemical effect of the increased water vapour on vortex average Antarctic ozone depletion in June through October was minor (less than 4 DU). Zhou et al. (2024) confirm this conclusion but find somewhat more ozone loss caused by the injected water vapour ($\approx$ 10 DU) at the vortex edge. The observation of a small impact of water vapour injected into the stratosphere on polar ozone loss is consistent with the notion put forward in this paper that low temperatures in the vortex, which occur regularly in the Antarctic, limit the atmospheric water vapour to the water vapour saturation pressure over ice and thus remove any anomalies through dehydration before they can affect ozone loss.

The impact of the stratospheric water vapour enhancement through the Hunga eruption on Antarctic ozone has further been assessed in the analysis of satellite observations (Santee et al., 2024). It was observed that the Hunga eruption increased the vertical extent of PSC formation and chlorine activation in early Austral winter in the Antarctic vortex in 2023 (the Antarctic season influenced most strongly by the Hunga eruption). Nonetheless, ozone depletion in the Antarctic in 2023 was unremarkable throughout the lower stratosphere (Santee et al., 2024).

The very minor impact of the huge water vapour injections into the stratosphere by the the Hunga volcano on Antarctic ozone in the 2023 season (Wohltmann et al., 2023; Fleming et al., 2024; Zhou et al., 2024; Santee et al., 2024) is consistent with the very small impact of initial water vapour in mid-winter and the subsequent formation of ice PSC particles in the model simulations presented here. First, the low temperatures in the lower stratosphere in the core of the Antarctic vortex determine mid-winter water vapour (independent of the amount of water vapour present at the time of the formation of the vortex). Second, even if higher water vapour mixing ratios prevailed in mid-winter, chlorine activation and chemical ozone loss remain practically unaltered (Fig. 4 of the submitted manuscript).

**Minor comments:**

**L17:** You want to add that the temperature range refers to potential temperature, the vertical coordinate in CLaMS.

Thanks, "potential temperature" has been added.

**L23:** Replace "although" with "notwithstanding" Done.

**L60:** Conventional wisdom has it that NAT is important here too. Please comment. I suggest to replace "ice particles" with "PSC particles"

We agree with this comment. We changed the wording and say PSC particles now; here is the new text: "Heterogeneous chlorine activation, enhanced concentrations of active chlorine and subsequent ozone loss occur frequently in the polar regions. Under exceptional circumstances chlorine activation also occurs in the mid-latitudes for conditions of low temperatures and enhanced water vapour. The surfaces for heterogeneous reactions might be provided for example by stratospheric PSC particles, stratospheric sulphate aerosol particles (potentially enhanced by volcanic eruptions or climate intervention) or by wildfire smoke injected into the stratosphere..."

**L116:** Replace "on" with "to" Done.

**References**

Fleming, E. L., Newman, P. A., Liang, Q., and Oman, L. D.: Stratospheric temperature and ozone impacts of the Hunga Tonga-Hunga Ha'apai water vapor injection, J. Geophys. Res., 129, e2023JD039298, URL `https://doi.org/10.1029/2023JD039298`, 2024.

Santee, M., Manney, G., Lambert, A., Millan, L., Livesey, N., Pitts, M., Froidevaux, L., Read, W., and Fuller, R.: The Influence of Stratospheric Hydration from the Hunga Eruption on Chemical Processing in the 2023 Antarctic Vortex, ESS Open Archive, https://doi.org/10.22541/essoar.170542085.55151307/v1, 2024.

Wohltmann, I., Santee, M. L., Manney, G. L., and Millán, L. F.: The chemical effect of increased water vapor from the Hunga Tonga-Hunga

Ha'apai eruption on the Antarctic ozone hole, Geophys. Res. Lett., 51, e2023GL106980, URL https://doi.org/10.1029/2023GL106980, 2023.

Zhou, X., Dhomse, S. S., Feng, W., Mann, G., Heddell, S., Pumphrey, H., Kerridge, B. J., Latter, B., Siddans, R., Ventress, L., Querel, R., Smale, P., Asher, E., Hall, E. G., Bekki, S., and Chipperfield, M. P.: Antarctic Vortex Dehydration in 2023 as a Substantial Removal Pathway for Hunga Tonga-Hunga Ha'apai Water Vapor, Geophys. Res. Lett., 51, e2023GL107630 2023GL107630, https://doi.org/https://doi.org/10.1029/2023GL107630, 2024.

---

## Author Comment (AC2)

*Reply to comments by reviewer two on*

**The impact of dehydration and extremely low HCl values in the Antarctic stratospheric vortex in mid-winter on ozone loss in spring**

*by Yiran Zhang-Liu et al.*

We thank the reviewer very much for her/his interest in our paper and for very helpful comments. The comments are repeated below in blue and a point-by-point response is given in normal font and black colour.

The paper has been revised in view of the comments in all three reviews.

**Review two**

General comments

The manuscript describes new modelling simulations of Antarctic ozone depletion using the well-regarded CLaMS model with meteorological fields from ECMWF.

Building on previous work by much the same team, the same techniques as previously used are again made use of to study the effects of making specific improvements to a number of the parameters of the simulation, namely updated reaction rates and more realistic values for water vapour and HCl as seen in observations. The authors find that with these changes, the model still simulates extremely low ozone in late September, as required to match observations.

While this could be seen as a null result which doesn't add very much to our understanding of polar ozone depletion, it is good science to investigate the effect of all such potential issues in previous work and to assess the sensitivities of the earlier results.

The subject matter of polar ozone depletion is central to the scope of ACP and I believe the manuscript is suitable for publication after some fairly easy revisions.

Thank you very much for these comments on our paper. Indeed, we agree that it constitutes "good science" to investigate the effect of potential issues in previous work. All comments in the review have been taken into account

and a revised version of the paper has been created.

My only major concern is that, while I find the manuscript is very clearly written, in the sense that each individual sentence is well-written and easily understood, the broader narrative is not very clearly expressed.

The entire paper has been revised with this comment in mind. See below for changes in detail.

I would like to see several points being better discussed for the benefit of the reader.

The abstract and the introduction need to explain better that this work is building on previous results. Similarly, the core method of using a single reference trajectory to evaluate the model output needs to be discussed (the readers shouldn't have to refer to the older papers) and the strengths and weaknesses of this approach outlined.

This is a valid point. However, the abstract is restricted in length (according to ACP standards) so there are limits to what is possible in the abstract. Nonetheless, we state in the abstract now in response: "Simulations of Antarctic chlorine and ozone chemistry in previous work show that in the core of the Antarctic vortex (16–18 km, 85–55 hPa, 390–430 K) HCl null cycles (initiated by reactions of Cl with $CH_4$ and $CH_2O$) are effective."

In the introduction, there is more room for discussion; we state now:

"In the present study, we extend earlier work on HCl null cycles (Grooß et al., 2011; Müller et al., 2018; Zafar et al., 2018) investigating the chemical processes in the core of the Antarctic vortex in the lower stratosphere (16–18 km, 85–55 hPa, 390–430 K), where extremely low ozone molar mixing ratios in spring are reached regularly (Solomon et al., 2005; Johnson et al., 2023). [. . . ] The earlier work (Grooß et al., 2011; Müller et al., 2018) was based on a detailed examination of a single trajectory and an analysis of multi-trajectory simulations. Here we do *not* employ a three-dimensional model version [. . . ], which is based on global or hemispheric meteorological fields and includes atmospheric mixing (e.g., Poshyvailo et al., 2018; Grooß and Müller, 2021; Sonnabend et al., 2024)."

Further, in response to this review, there is also a more detailed discussion of the multi-trajectory analysis in section 3.4 now.

The authors don't explain why unrealistic choices of water vapour (in particular) and HCl were used in the older work – according to section 2.2.1, two 2018 papers used 4.1 ppm but the observations listed giving a lower

concentration were generally well known before 2018.

First, we agree with the comment that regarding initial water vapour; clearly the issue of Antarctic dehydration was known prior to 2018 (e.g., Kelly et al., 1989; Vömel et al., 1995; Nedoluha et al., 2002; Jiménez et al., 2006, and other references in the submitted manuscript). The initial value of 4.1 ppm chosen in previous work (Müller et al., 2018; Zafar et al., 2018) was too high for an entire Antarctic winter. Such a value is however appropriate for conditions on 1 June, but dehydration occurs thereafter in the Antarctic stratosphere (although this dehydration is not represented in the trajectory model employed here). In response to this comment, these arguments are now better discussed in section 2.2.1. In particular, this issue should be resolved here by showing the simulation with a lower (dehydrated) initial value for $H_2O$, which is close to observations. However, assuming somewhat higher initial water vapour mixing ratios does not seem completely irrelevant considering the case of the Honga volcanic eruption raised in review one. (And we have refrained from a discussion in the paper *why* particular choices were made in previous work).

Second, regarding initial HCl, this issue was not well established prior to 2018 (Wohltmann et al., 2017; Grooß et al., 2018). Indeed, the processes causing the low winter HCl are not yet established today. Overall, we argue that this paper makes an important point stating that neither the initial $H_2O$ nor the initial HCl have a significant impact on Antarctic chlorine chemistry and ozone depletion (corroborating the findings of earlier studies, Müller et al., 2018; Zafar et al., 2018).

The authors also don't suggest any other ways the new simulations could be tested other than the effect on the ozone concentration at the end of the reference trajectory – for example, wouldn't there be some observational consequences of the much greater surface area of ice clouds shown in figure 3?

We agree and have now added (in section 3.2) to the paper: "There should be observational consequences of the very different ice surfaces in simulations S2 and S3 (Fig. 3), i.e. observations should allow discriminating between the hypotheses about initial water vapour in simulations S2 and S3." Further (section 3.3.1), we have added now: "Differences in the temporal development of HOCl, $ClO_x$, and HCl between simulations S3 and S4 are greater than for ozone and thus might possibly be simpler to detect in satellite observation than a different temporal development of ozone."

Section 3.4 was not very clear in terms of the motivation, the details of the method or what exactly the results were showing, and needs some particular extra work.

We agree that the motivation for the multi-trajectory simulation in section 3.4 needs to be clearer. The point is that the reference trajectory is typical for the conditions in the vortex core in the lower stratosphere. We have added to following text to the paper: "In the discussions above, one particular (reference) trajectory was considered. Nonetheless, this trajectory is representative for the conditions in the core of the Antarctic vortex at 16–18 km (85–55 hPa, 390–430 K). To demonstrate this, we selected here twenty one trajectories passing the South Pole (in late September/early October) at the 400 K potential temperature level; these trajectories include diabatic descent and latitude variations."

Regarding the results, it is stated in the paper that "all trajectories show strongly enhanced values of $ClO_x$ over the period of strong ozone loss in August and September, consistent with suppressed values of HCl".

There is some repetition in the text which could be cleaned up.

Thanks for pointing this out – we have revised the text throughout the paper.

Thanks for this comment. We have also invested some effort in keeping the references correct and up-to date.

**Specific comments:**

Lines 2-11 Please re-write the abstract to better describe the purpose of the paper

The purpose of the paper is now clearer in the abstract: " Here we investigate the impact of the observed dehydration in Antarctica, [. . . ]; however the efficacy of HCl null cycles is not affected. Moreover, also when using the observed very low HCl molar mixing ratios in Antarctic winter as initial values; HCl null cycles are efficient in maintaining low HCl (and high $ClO_x$) throughout winter/spring." (see also comments above).

Line 52 Please state the reactions you are referring to here

Done, we state now: ". . . in spite of increasingly rapid formation of HCl in the gas phase (through the reactions of Cl with $CH_4$ and $CH_2O$ Müller

et al., 2018)."

Done, we state now: "(which occurs on the surfaces of nitric acid trihydrate (NAT) and ice particles or within supercooled (liquid) ternary solutions and cold liquid aerosol particles)"

Lines 56-57 please state or refer to the specific reactions

Done, we have added now: "... through the reactions of Cl with $CH_4$ and $CH_2O$ ..."

Line 75-76 "the uptake of $HNO_3$ on ice particles" has not been mentioned until now

We agree and have provided more background here (including a new citation). The text in the manuscript reads now: "Ice clouds are very efficient in sequestering $HNO_3$ from the gas-phase (e.g. Hynes et al., 2002), thus a lower occurrence of ice clouds in the model reduces substantially the uptake of gas-phase $HNO_3$ on ice particles".

Line 87 This reaction has not been previously mentioned either

We agree, we have removed the mentioning of this reaction $(ClO + CH_3O_2)$ in line 87. The reaction is now discussed in the introduction below reaction R2.

Lines 98-101 How does the model know what the surface areas are of these different types of clouds though?

Temperature is key here. We have extended the model description to address this point. In particular, we state in the paper now: "NAT particles are assumed to form at a supersaturation of 10 from liquid ternary solutions or from ice evaporation. Ice is formed in the model at the equilibrium temperature (no supersaturation). The initial density of liquid (binary) aerosol particles is assumed to be 10 $cm^{-3}$. The condensable material for liquid ternary particles, NAT and ice is determined from the equilibrium with the gas-phase."

Table 1 – how have these values been determined though? In particular, how have you decided the Bromine concentration?

We agree that more information on the initial conditions should be given in the paper. In response, we have added to following text: "The values for key species at the start of the simulation on 1 May 2003 at 430 K potential temperature for $O_3$, $HNO_3$, and $N_2O$ are based on MIPAS-Envisat observations; further, tracer correlations of $N_2O$ with $Cl_y$, $Br_y$, and $NO_y$ were employed (see Grooß et al., 2011, for details). With the exception of $H_2O$, the initial values used here are identical to those used in earlier work (Müller et al., 2018; Zafar et al., 2018)."

Lines 120-139 The reader is left to puzzle why the 2018 papers used 4.1 ppm when there was such an abundance of observational data available to support a lower value – this point should be briefly discussed, otherwise it sounds strange.

We agree that adding some discussion is warranted here, which was done (see above). We also agree with the reviewer that the process of Antarctic dehydration was established prior to 2018 (e.g., Kelly et al., 1989; Vömel et al., 1995; Nedoluha et al., 2002; Jiménez et al., 2006, and other references in the submitted manuscript). Clearly, the initial value of 4.1 ppm chosen in previous work (Müller et al., 2018; Zafar et al., 2018) was too high for an entire Antarctic winter. (As is stated in the paper now, section 2.2.1). This issue is resolved here by showing the simulation with a lower (dehydrated) initial value for $H_2O$, which is close to observations. However, assuming somewhat higher initial water vapour mixing ratios does not seem completely irrelevant considering the case of the Honga volcanic eruption raised in review one. (And we have refrained from a discussion in the paper *why* particular choices were made in previous work).

Line 148 – I don't think you quite mean "it must be a process missing in the models". You next state it could be a temperature bias in the meteorological fields, which isn't a missing process in the models.

Yes, this is correct; we have removed "(but it must be a process missing in the models)" from the manuscript.

Line 220 What do you mean by "the occurrence heterogeneous reactions"?

We agree that our formulation was not good. We are more explicit now and say in the paper: "This is consistent with the notion that the *rate constant* of the heterogeneous reactions within HCl null cycles is of little relevance for the efficacy of the HCl null cycles. Although it is important for the efficacy of the HCl null cycles that temperatures are sufficiently low so that particles are present and heterogeneous reactions occur..."

Figure 2 – A general question about the method – the trajectory is calculated for months but its path could not possibly be accurately determined for such a long period of time – does this matter?

It does not matter. However, it is correct pointing out that a trajectory cannot be accurately determined for a long period of time. Nonetheless, accurately following an air parcel is also not possible in a classic three-dimensional model (e.g., excessive mixing across transport barriers or comparisons over several months on a pressure surface). However, following an air parcel (as done here) allows a pathway analysis to be conducted, which is more difficult in a three-dimensional framework. An important point is that the trajectory discussed in the main part the results section (independent of the details of the trajectory) is representative for the conditions in the core of the vortex in the lower stratosphere (see the point below).

Lines 225-271 Section 3.4 is not explained well enough, you need to better motivate this section for the reader, explain what exactly the different trajectories are, and discuss what it shows.

We agree, section 3.4 (in particular the motivation) has been changed, see also response on this point above. Further the paper states regarding this issue: "As in the reference simulation, the period of rapid ozone loss [...] ends abruptly with chlorine deactivation through very rapid formation of HCl [...]. After deactivation, HCl values remain high and practically unchanged in the box model simulation". We are confident that section 3.4 is now better motivated and explained.

Lines 296-298 This sentence reads very awkwardly at the moment and needs some minor re-wording. " ...while ozone depletion is somewhat enhanced ...ozone depletion is not strongly affected"

We agree – the sentence has been changed; we dropped "while ozone depletion is somewhat enhanced under these conditions"; the sentence now reads "...ozone depletion is not strongly affected by the initial values of HCl..."

**References**

Grooß, J.-U. and Müller, R.: Simulation of record Arctic stratospheric ozone depletion in 2020, J. Geophys. Res., 126, e2020JD033339, https://doi.org/10.1029/2020JD033339, 2021.

Grooß, J.-U., Brautzsch, K., Pommrich, R., Solomon, S., and Müller, R.: Stratospheric ozone chemistry in the Antarctic: What controls the lowest

values that can be reached and their recovery?, Atmos. Chem. Phys., 11, 12 217–12 226, https://doi.org/10.5194/acp-11-12217-2011, 2011.

Grooß, J.-U., Müller, R., Spang, R., Tritscher, I., Wegner, T., Chipperfield, M. P., Feng, W., Kinnison, D. E., and Madronich, S.: On the discrepancy of HCl processing in the core of the wintertime polar vortices, Atmos. Chem. Phys., pp. 8647–8666, https://doi.org/10.5194/acp-18-8647-2018, 2018.

Hynes, R. G., Fernandez, M. A., and Cox, R. A.: Uptake of $HNO_3$ on water-ice and coadsorption of $HNO_3$ and HCl in the temperature range 210–235 K, J. Geophys. Res., 107, 4797, https://doi.org/https://doi.org/10.1029/2001JD001557, 2002.

Jiménez, C., Pumphrey, H. C., MacKenzie, I. A., Manney, G. L., Santee, M. L., Schwartz, M. J., Harwood, R. S., and Waters, J. W.: EOS MLS observations of dehydration in the 2004–2005 polar winters, Geophys. Res. Lett., 33, L16806, https://doi.org/10.1029/2006GL025926, 2006.

Johnson, B. J., Cullis, P., Booth, J., Petropavlovskikh, I., McConville, G., Hassler, B., Morris, G. A., Sterling, C., and Oltmans, S.: South Pole Station ozonesondes: variability and trends in the springtime Antarctic ozone hole 1986–2021, Atmos. Chem. Phys., 23, 3133–3146, https://doi.org/10.5194/acp-23-3133-2023, 2023.

Kelly, K. K., Tuck, A. F., Murphy, D. M., Proffitt, M. H., Fahey, D. W., Jones, R. L., McKenna, D. S., Loewenstein, M., Podolske, J. R., Strahan, S. E., Ferry and K. R. Chan and J. F. Vedder, G. V., Gregory, G. L., Hypes, W. D., McCormick, M. P., Browell, E. V., and Heidt, L. E.: Dehydration in the lower Antarctic stratosphere during late winter and early spring, 1987, J. Geophys. Res., 94, 11 317–11 357, 1989.

Müller, R., Grooß, J.-U., Zafar, A. M., Robrecht, S., and Lehmann, R.: The maintenance of elevated active chlorine levels in the Antarctic lower stratosphere through HCl null cycles, Atmos. Chem. Phys., 18, 2985–2997, https://doi.org/10.5194/acp-18-2985-2018, 2018.

Nedoluha, G. E., Bevilacqua, R. M., and Hoppel, K. W.: POAM III measurements of dehydration in the Antarctic and comparison with the Arctic, J. Geophys. Res., 107, 2002.

Poshyvailo, L., Müller, R., Konopka, P., Günther, G., Riese, M., Podglajen, A., and Ploeger, F.: Sensitivities of modelled water vapour in the lower stratosphere: temperature uncertainty, effects of horizontal transport and small-scale mixing, Atmos. Chem. Phys., 18, 8505–8527, https://doi.org/10.5194/acp-18-8505-2018, 2018.

Solomon, S., Portmann, R. W., Sasaki, T., Hofmann, D. J., and Thompson, D. W. J.: Four decades of ozonesonde measurements over Antarctica, J. Geophys. Res., 110, D21311, https://doi.org/10.1029/2005JD005917, 2005.

Sonnabend, J., Grooß, J.-U., Ploeger, F., Hoffmann, L., Jöckel, P., Kern, B., and Müller, R.: Lagrangian transport based on the winds of the icosahedral nonhydrostatic model (ICON), Meteorol. Z., accepted, 2024.

Vömel, H., Oltmans, S. J., Hofmann, D. J., Deshler, T., and Rosen, J. M.: The evolution of the dehydration in the Antarctic stratospheric vortex, Geophys. Res. Lett., 100, 13 919 – 13 926, 1995.

Wohltmann, I., Lehmann, R., and Rex, M.: A quantitative analysis of the reactions involved in stratospheric ozone depletion in the polar vortex core, Atmos. Chem. Phys., 17, 10 535–10 563, https://doi.org/10.5194/acp-17-10535-2017, 2017.

Zafar, A. M., Müller, R., Grooß, J.-U., Robrecht, S., Vogel, B., and Lehmann, R.: The relevance of reactions of the methyl peroxy radical ($CH_3O_2$) and methylhypochlorite ($CH_3OCl$) for Antarctic chlorine activation and ozone loss, Tellus B: Chemical and Physical Meteorology, 70, 1–18, https://doi.org/10.1080/16000889.2018.1507391, 2018.

---

## Author Comment (AC3)

*Reply to comments by reviewer three on*

**The impact of dehydration and extremely low HCl values in the Antarctic stratospheric vortex in mid-winter on ozone loss in spring**

*by Yiran Zhang-Liu et al.*

We thank the reviewer very much for her/his interest in our paper and for very helpful comments. The comments are repeated below in blue and a point-by-point response is given in normal font and black colour.

The paper has been revised in view of the comments in all three reviews.

**Review three**

**General comments**

The paper entitled: "The impact of dehydration and extremely low HCl values in the Antarctic stratospheric vortex in mid-winter on ozone loss in spring" explores in depth the role of HCL null cycles on maintaining active chlorine in early Antarctic spring. The paper looks into the roles of initial wintertime HCL concentrations (where there is a known discrepancy between models and observations), a correction to the CLO + CH3O2 reaction rate, and dehydration on the HCL null cycles.

Overall, the paper is well written and is a nice addition to literature on Antarctic chlorine partitioning. Further knowledge on the HCL null cycles and the factors that affect them is an important and welcome advancement to the knowledge of Antarctic chlorine partitioning and ozone loss. I have a few comments below that I would like to see addressed. I suggest publication after the following minor revisions.

Thank you very much for these comments on our paper. All comments have been taken into account and a revised version of the paper has been created; in particular, we have restructured the "incorrect A-factor analysis" substantially as suggested.

**Main comments**

The authors present the majority of the results in a concise way, however I found the discussion around the CLO+CH3O2 reaction rates, specifically

discussion of results of the incorrect A-factor analysis in the methods section, hard to follow. I feel this section can be shortened somewhat or made more concise, especially as results are discussed here but not shown (apart from a few values printed in text). The authors also state in the abstract that there is little difference between the two simulations when using the old (Sander) rates and new (Burkholder) rates. Looking at Figure 2 it looks to me that the differences can be quite significant between the two simulations and remains through to December 1. This may seem insignificant, but such differences after only one reaction is notable. This conclusion is a theme in the other cases investigated as well.

We agree with this comment. In response, we have restructured section 2.3.2; we think that the "incorrect A-factor" analysis is much clearer now. We have also added information of the unit of the rate constant $k$. Further, we agree that the model results (for simulation S4) with the incorrect A-factor should not be discussed in section 2.3.2; we have moved this part to section 3.3 below. There is now a better and clearer discussion of the issue and the error in Eqns. 2 and 3 has been corrected (see also below).

Finally, the review is correct in pointing out that there is the problem that the results on the reaction $ClO + CH_3O_2$ are discussed in the manuscript *but not shown* – this has been changed. The newly added figure 6 now shows the results on the reaction $ClO + CH_3O_2$, so that the accompanying text is much easier to follow.

Is there no role of CLONO2+HCL in spring in maintaining elevated active chlorine? Your box model clearly shows no CLONO2 at all through to December. However, I believe there should be some elevated CLONO2 when spring arrives and therefore this reaction should also play some role. For example Solomon et al. (2015) Figure 3 shows elevated springtime CLONO2 levels from MIPAS observations. The reaction is likely not proceeding as fast as HOCL+HCL, but will the addition of this reaction affect the null cycles in any way? I feel this needs to be at least addressed in the paper.

It is important to point out that the reaction $ClONO_2 + HCl$ was *not* taken out of the system of reactions when the HCl null cycles were identified (Müller et al., 2018). Quoting from Müller et al. (2018) regarding the pathway analysis "As input it [the pathway analysis] requires a set of chemical reaction equations and reaction rates, which are usually provided by a chemical model. Starting from the individual reactions (and their rates) as initial pathways, longer pathways are constructed step by step by connecting shorter

ones. [...] A rate for each pathway is calculated" (see Lehmann, 2004, for more details). Thus, the HCl null cycles (cycles C1 and C2) have emerged in the pathway analysis (Lehmann, 2004) from the complete chemical set of reactions.

Further, Figure 3 of Solomon et al. (2015) is for 61 hPa, that is for somewhat higher altitudes than studied here (roughly 75-80 hPa for the period of strongest ozone depletion). Also, for days 220 to 250 (Fig. 3), the observed values of $ClONO_2$ (MIPAS) are extremely low (similar as in the simulations presented here).

But in any case, we agree with the review that more discussion is necessary; in response we have added the following text to the introduction of the paper.:

"However, at altitudes somewhat greater than 18 km (55 hPa, 430 K) and for conditions in the lower stratosphere closer to the edge of the polar vortex, $HNO_3$ will not continuously be sequestered in PSCs, so that periods with enhanced gas-phase concentrations of $HNO_3$ (compared to the vortex core) will occur. Under such conditions, more $NO_2$ will be available in the gas-phase (e.g., de Laat et al., 2024), enhancing the production of $ClONO_2$, so that reaction R1 will have a much stronger impact on chlorine chemistry. As a result, the chemistry of HCl null cycles will be more complex."

**Specific comments**

Lines 53-55. Does CLONO2+HCL also play a role in maintaining elevated chlorine? See more extensive discussion above. We agree and have added the following text: "However, at altitudes somewhat greater than $\approx$ 18 km (55 hPa, 430 K) and for conditions in the lower stratosphere closer to the edge of the polar vortex, $HNO_3$ will not continuously be sequestered in PSCs, so that periods with enhanced gas-phase concentrations of $HNO_3$ (compared to the vortex core) will occur. Under such conditions, more $NO_2$ will be available in the gas-phase (e.g., de Laat et al., 2024), enhancing the production of $ClONO_2$, so that reaction R1 will have a much stronger impact on chlorine chemistry. As a result, the chemistry of HCl null cycles will be more complex." (see also above).

Lines 179 and 182. I believe these equations should be A*exp(-E/RT) not A*exp(-R/ET)? Based on Figure 1 and Table 3 it looks like this is just a typo, but please check. The reviewer is right, Eqns. 2 and 3 in the submitted

version were not correct. The error has been corrected and the Arrhenius equation is now better explained, so that there should not be any remaining misunderstandings. Figure 1 and Table 3 were calculated using the correct formula. Thanks very much for catching this.

Section 3.1. I would like to see this section expanded on a little to explain why there are differences when changing from the older to newer rate recommendations.

The difference between simulations S1 and S2 are minor (most visible in the bottom plot with ozone on a logarithmic scale, see Fig. 2); However, we have added a sentence on the somewhat different development of $ClONO_2$ in simulations S1 and S2 to the discussion here. Canty et al. (2016) have already provided a detailed discussion of updates to JPL recommendations by Burkholder et al. (2015). In response to the comment we state now in the paper: "...(although there are a few periods with more $ClONO_2$ for an initial water vapour molar mixing ratio of 4.11 ppm, Fig. 2). This has consequences for chemical ozone depletion (Fig. 2). There is a slightly lower minimum value of ozone ($\approx 10$ ppb lower) for an initial water vapour molar mixing ratio of 4.11 ppm."

Line 225. The authors state: "Further, a substantial difference in initial water vapour mixing ratios does not result in a substantial difference of polar chlorine chemistry and ozone loss (Fig. 4). There is a slightly lower minimum value of ozone ($\approx 10$ ppb lower) for an initial water vapour mixing ratio of 4.11 ppm." Again this seems a quite significant change to me. Some discussion of why this isn't would be welcome here.

We agree that more discussion is warranted here. We have reworked the entire section. The main point is that the HCl null cycles do not depend on the actual rate constant of the heterogeneous reaction $HOCl + HCl \longrightarrow$ ; in other words $HOCl + HCl \longrightarrow$ is still "fast enough" independent of the initial water vapour. We state now in the paper: " The rate constant of heterogeneous reactions is influenced strongly by the type of the available PSC particles. The efficacy of the HCl null cycles, however, is limited by the rates of the reactions of Cl with $CH_4$ and $CH_2O$ [...]. The HCl null cycles are relevant for the maintenance of high levels of active chlorine (Müller et al., 2018)". In response to review one, we have also added a section on the eruption of the Honga volcano, where a further discussion of initial water vapour is provided.

Line 240-245. The earlier onset of ozone loss here is interesting and I

would like to see it discussed more. This to me is quite substantial especially when early winter HCL conditions is something that fully coupled models can't simulate accurately at the moment, as you mention in the paper.

Again, we agree that more discussion is warranted here. We have revised the relevant section of the manuscript. In particularly, we have added to the text: "However, there is clearly an earlier onset of ozone depletion when $HCl_{initial} = 0$ is employed, with the difference between simulation S3 and S4 notable in late August/early September."

**Technical corrections**

Line 193. Please remove "a" from "to a much larger value". Thanks, sentence is reformulated.

Line 295. Suggest rewording "Further, while ozone depletion is somewhat enhanced under these conditions, ozone depletion is not strongly affected" as it currently sounds contradictory. Thanks, the sentence is simpler and clearer now: "Further, ozone depletion is not strongly affected by the initial values of HCl (and also the minimum values of Antarctic ozone reached are similar) consistent with Grooß et al. (2018)."

**References**

Burkholder, J. B., Sander, S. P., Abbatt, J. P. D., Barker, J. R., Huie, R. E., Kolb, C. E., Kurylo, M. J., Orkin, V. L., Wilmouth, D. M., and Wine, P. H.: Chemical kinetics and photochemical data for use in atmospheric studies, Evaluation Number 18, JPL Publication 15-10, 2015.

Canty, T. P., Salawitch, R. J., and Wilmouth, D. M.: The kinetics of the ClOOCl catalytic cycle, J. Geophys. Res., 121, 13 768–13 783, https://doi.org/10.1002/2016JD025710, 2016.

de Laat, A., van Geffen, J., Stammes, P., van der A, R., Eskes, H., and Veefkind, J. P.: The Antarctic stratospheric nitrogen hole: Southern Hemisphere and Antarctic springtime total nitrogen dioxide and total ozone variability as observed by Sentinel-5p TROPOMI, Atmos. Chem. Phys., 24, 4511–4535, https://doi.org/10.5194/acp-24-4511-2024, 2024.

Grooß, J.-U., Müller, R., Spang, R., Tritscher, I., Wegner, T., Chipperfield,

M. P., Feng, W., Kinnison, D. E., and Madronich, S.: On the discrepancy of HCl processing in the core of the wintertime polar vortices, Atmos. Chem. Phys., pp. 8647–8666, https://doi.org/10.5194/acp-18-8647-2018, 2018.

Lehmann, R.: An algorithm for the determination of all significant pathways in chemical reaction systems, J. Atmos. Chem., 47, 45–78, 2004.

Müller, R., Grooß, J.-U., Zafar, A. M., Robrecht, S., and Lehmann, R.: The maintenance of elevated active chlorine levels in the Antarctic lower stratosphere through HCl null cycles, Atmos. Chem. Phys., 18, 2985–2997, https://doi.org/10.5194/acp-18-2985-2018, 2018.

Solomon, S., Kinnison, D., Bandoro, J., and Garcia, R.: Simulation of polar ozone depletion: An update, J. Geophys. Res., 120, 7958–7974, https://doi.org/10.1002/2015JD023365, 2015.

---

## Author Response (AR3)

*Reply to comments by reviewer two (second round) on*

**The impact of dehydration and extremely low HCl values in the Antarctic stratospheric vortex in mid-winter on ozone loss in spring**

*by Yiran Zhang-Liu et al.*

We thank the reviewer again for her/his interest in our paper and for very helpful additional comments. The comments are repeated below in blue and a point-by-point response is given in normal font and black colour.

The paper has been revised in view of the comments in this second round of comments.

**Review two (second round)**

General comments

I would like to thank the authors for addressing my previous comments. I think the paper has been improved from the previous version. I am happy for this paper to be accepted after the following minor comments have been addressed.

Thank you!

Following my previous comment about ClONO2 + HCL reaction, the author's response states: "Further, Figure 3 of Solomon et al. (2015) is for 61 hPa, that is for some- what higher altitudes than studied here (roughly 75-80 hPa for the period of strongest ozone depletion). Also, for days 220 to 250 (Fig. 3), the observed values of ClONO2 (MIPAS) are extremely low (similar as in the simulations presented here)." I appreciate the author's reasoning regarding the ClONO2 + HCl reaction not being as important at lower altitudes for days 220-250, However, the null cycles presented in the paper seen to show active chlorine maintenance through to the end of September ($\sim$ day 275) in their Figures 4 and 5. I suspect by the end of September ClONO2 concentrations will be greater than HOCl even at lower altitudes (although that is clearly not shown in your box model, so I could be wrong)? I think a little more expansion of their explanation (which I agree with for earlier time periods at lower altitudes) needs to be made for

the later time period too. I would be nice to have some confirmation using observations: ACE-FTS has measurements of ClONO2 that extent as low as ~10 km at high Southern latitudes during August and September that could be used to confirm that ClONO2 is or is not important during the later time period.

Thanks for these remarks. First, we should point out that a follow up study is planned, which will address the $ClONO_2$ issue in more detail. However, clearly, this paper needs to be complete. For the conditions considered in this paper, the temporal development of both HOCl and $ClONO_2$ is shown for the entire period (including end of September; $\sim$ day 275, see e.g., Fig. 2 but also Figs. 4 and 5, panels c and d). At the end of September (for the conditions considered here), $ClONO_2$ is almost zero, while HOCl is about 0.1 ppb (or more) as long as $ClO_x$ is elevated.

However this issue needs to be better explained/discussed, so we have added to the text (at the end of section 3.3.1): "Under the conditions discussed here, values of $ClONO_2$ remain strongly depressed (close to zero, with few exceptions; Fig. 5, panel d). This statement is true for the entire simulated period, including the period of strongest ozone loss throughout September. This observation is consistent with the dominance of the HCl null cycles that do not involve the heterogeneous reaction R1 (Müller et al., 2018, see also appendix A)."

Line 54. The author's state "so that there is no full activation in this step". There is still complete activation though I believe? We just don't understand why it is occurring? It's likely still due to reaction R1. Or do you mean only in the model?

The reviewer is correct here. We mean "in the model" and this is clarified by saying explicitly in the paper now: "In the Antarctic lower stratosphere, the initial concentrations of HCl are greater than those of $ClONO_2$ (Jaeglé et al., 1997; Santee et al., 2008; Nakajima et al., 2020). Thus, in the absence of chemical processes leading to a further loss in HCl, there is no full activation in this step. Such a behaviour is found in models (Grooß et al., 2018)."

Line 379-381 and Lines 405-409. The author's state: "Further, ozone depletion is not strongly affected by the initial values of HCl (and also the minimum values of Antarctic ozone reached are similar) consistent with Grooß et al. (2018)." Figure 5 to me shows that ozone depletion is strongly affected, maybe the minimum values are similar, but the earlier onset when HCl = 0 looks like a large difference to me. I thank the authors for including discussion of this in the relevant section, but I think it should also be mentioned here and in the conclusions. Something like "Initial values of HCl are seen to impact the timing of onset of ozone depletion and the timing of maximum ozone loss, but don't significantly impact the minimum values."

We agree. In response, we have changed the text in the discussion: "Further, ozone depletion is affected by the initial values of HCl, namely the timing of maximum ozone loss. However, the minimum values of Antarctic ozone reached are similar, consistent with Grooß et al. (2018)" and in the conclusions: "Further, $ClO_x$ is enhanced throughout winter and spring, and HCl molar mixing ratios remain very low until rapid chlorine deactivation occurs into HCl. Also the strength of the ozone loss rate and the timing of maximum ozone loss is affected by the initial value of HCl, but not the minimum ozone value (consistent with  Grooß et al., 2018); the simulated ozone minimum values differ by $\approx 10$ ppb".

Line 388-390: Does this sentence: "Second, the HCl null cycles require the heterogeneous reaction R2 ($HCl + HOCl \longrightarrow Cl_2 + H_2O$) to proceed at a substantial rate" contradict the earlier statement on Line 258-259: "This finding is consistent with the notion that the rate constant of the heterogeneous reactions within HCl null cycles is of little relevance for the efficacy of the HCl null cycles"?

Thanks for pointing this out. Indeed "substantial" is not correct here. In the revised version of the paper it is stated now: "Second, the HCl null cycles require the heterogeneous reaction R2 ($HCl + HOCl \rightarrow Cl_2 + H_2O$) to proceed; i.e., temperatures need to be sufficiently low."

**References**

Grooß, J.-U., Müller, R., Spang, R., Tritscher, I., Wegner, T., Chipperfield, M. P., Feng, W., Kinnison, D. E., and Madronich, S.: On the discrepancy of HCl processing in the core of the wintertime polar vortices, Atmos. Chem. Phys., pp. 8647–8666, https://doi.org/10.5194/acp-18-8647-2018, 2018.

Jaeglé, L., Webster, C. R., May, R. D., Scott, D. C., Stimpfle, R. M., Kohn, D. W., Wennberg, P. O., Hanisco, T. F., Cohen, R. C., Proffitt, M. H., Kelly, K. K., Elkins, J., Baumgardner, D., Dye, J. E., Wilson, J. C., Pueschel, R. F., Chan, K. R., Salawitch, R. J., Tuck, A. F., Hovde, S. J.,

and Yung, Y. L.: Evolution and stoichiometry of heterogeneous processing in the Antarctic stratosphere,, J. Geophys. Res., 102, 13 235–13 253, https://doi.org/10.1029/97JD00935, 1997.

Müller, R., Grooß, J.-U., Zafar, A. M., Robrecht, S., and Lehmann, R.: The maintenance of elevated active chlorine levels in the Antarctic lower stratosphere through HCl null cycles, Atmos. Chem. Phys., 18, 2985–2997, https://doi.org/10.5194/acp-18-2985-2018, 2018.

Nakajima, H., Murata, I., Nagahama, Y., Akiyoshi, H., Saeki, K., Kinase, T., Takeda, M., Tomikawa, Y., Dupuy, E., and Jones, N. B.: Chlorine partitioning near the polar vortex edge observed with ground-based FTIR and satellites at Syowa Station, Antarctica, in 2007 and 2011, Atmos. Chem. Phys., 20, 1043–1074, https://doi.org/10.5194/acp-20-1043-2020, 2020.

Santee, M. L., MacKenzie, I. A., Manney, G. L., Chipperfield, M. P., Bernath, P. F., Walker, K. A., Boone, C. D., Froidevaux, L., Livesey, N. J., and Waters, J. W.: A study of stratospheric chlorine partitioning based on new satellite measurements and modeling, J. Geophys. Res., 113, D12307, https://doi.org/10.1029/2007JD009057, 2008.